# SPATIAL-TEMPORAL-SPECTRAL UNIFIED MODELING FOR REMOTE SENSING DENSE PREDICTION

## ABSTRACT

Despite advances in deep learning for dense prediction fueled by multi-source remote sensing data, a unified framework for handling heterogeneous inputs, multiple tasks, and open semantic categories remains elusive. Concretely, multiple correlated dense prediction tasks (e.g., building extraction, change detection) are often modeled independently, requiring separate networks or task-specific decoder heads. Moreover, the semantic label space is fixed during training; any modification to the class set requires full retraining, hindering adaptability in dynamic operational environments. Here, we introduce the Spatial-Temporal-Spectral Unified Network (STSUN), a unified framework for remote sensing dense prediction. The model employs a metadata-informed unified representation generated via hypernetworks, enabling adaptation to inputs with variable spatial resolutions, temporal lengths, and spectral band configurations. Furthermore, it unifies diverse dense prediction tasks within a single architecture through learnable task embeddings and semantic category embeddings that support arbitrary class subsets. Extensive experiments on datasets with diverse spatial–temporal–spectral configurations demonstrate the effectiveness of the proposed method, achieving unified modeling of multiple dense prediction tasks and semantic class predictions within a single model.

## 1 INTRODUCTION

With the continuous advancement of remote sensing technologies and the increasing diversity of data acquisition methods, the field of remote sensing has entered a phase of rapid development. Large-scale, multi-source remote sensing data has been widely applied to various dense prediction tasks, playing a crucial role in applications such as urban expansion monitoring (Benedek et al., 2012), land cover classification (Gu et al., 2022), and disaster damage assessment (Voigt et al., 2016). Remote sensing imagery and labels exhibit high heterogeneity across three key dimensions including spatial, temporal and spectral dimensions in dense prediction tasks, posing significant challenges for unified processing due to variations in image size, temporal length and spectral bands of input and output in practical applications (Reichstein et al., 2019).

Remote sensing dense prediction aims to assign pixel-wise labels to satellite imagery (Zhu et al., 2017), encompassing three primary tasks: semantic segmentation, binary change detection (BCD), and semantic change detection (SCD). Formally, let the dense prediction model be defined as a mapping function $\mathcal{F} : \mathbf{X} \to \mathbf{Y}$, where the input $\mathbf{X}$ and output $\mathbf{Y}$ are tensors defined as:

$$\mathbf{X} \in \mathbb{R}^{T_1 \times C_1 \times H_1 \times W_1}, \quad \mathbf{Y} \in \mathbb{R}^{T_2 \times C_2 \times H_2 \times W_2} \tag{1}$$

Here, $T$, $C$, $H$, and $W$ denote the temporal length, channel (or class) count, height, and width, respectively, assuming spatial consistency where $(H_1, W_1) = (H_2, W_2) = (H, W)$. Consequently, the specific dense prediction task is determined by the constraints imposed on the input-output temporal $(T_1, T_2)$ and spectral $(C_2)$ relationships:

$$\begin{cases} \text{Semantic Segmentation:} & T_2 = 1, \quad C_2 \geq 2 \\ \text{Binary Change Detection:} & T_2 = T_1 - 1, \quad C_2 = 2 \\ \text{Semantic Change Detection:} & T_2 = T_1, \quad C_2 \geq 2 \end{cases} \tag{2}$$

Physically, the input dimensions $T_1$, $C_1$, and $(H_1, W_1)$ explicitly parameterize the temporal coverage/resolution, data modality, and geographical coverage/spatial resolution, respectively. The output

dimensions $T_2$ and $C_2$ related to the task type and semantic category set, respectively. The relationship between dimensions and data-task-category underscoring the necessity for a unified modeling approach across spatial-temporal-spectral dimensions.

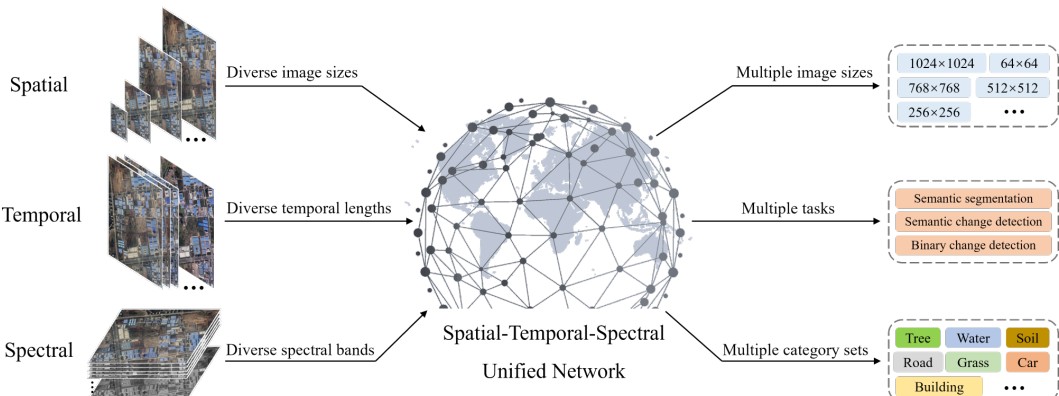

Figure 1: Illustration of Spatial-Temporal-Spectral Unified Network. STSUN is capable of handling input and output with arbitrary spatial, temporal and spectral dimension configurations, unifies semantic segmentation, binary change detection and semantic change detection tasks with flexible category sets.

While deep learning methods have achieved significant progress in remote sensing dense prediction, current architectures exhibit several limitations. **1. Fixed Configurations**: These models are typically designed for fixed input-output configurations, defined by specific image sizes $(H_1, W_1)$ and $(H_2, W_2)$), temporal lengths $(T_1, T_2)$ and spectral bands $(C_1, C_2)$. This rigidity restricts their adaptability to the diverse and heterogeneous remote sensing datasets encountered in practical applications, such as diverse satellite sensor data with varying spectral bands or multi-temporal imagery sequences with varying temporal lengths. **2. Fixed Task:** These models fail to leverage the inherent correlations among semantic segmentation, binary change detection, and semantic change detection tasks. This oversight often requires the development of distinct models or dedicated task-specific decoders for each task, hindering unified processing pipelines. **3. Fixed Category Set:** Existing approaches are generally constrained to dense prediction with a fixed set of output categories. Consequently, any alteration to the target dense prediction schema, even a slight change, often results in substantial performance degradation or complete incompatibility, necessitating extensive retraining or fine-tuning, which incurs considerable computational and temporal overhead.

To overcome these limitations, this study introduces the Spatial-Temporal-Spectral Unified Network (STSUN), as illustrated in Figure 1. STSUN offers a novel framework for unified representation and modeling of remote sensing data across diverse spatial, temporal and spectral dimensions, and addresses the aforementioned issues as follows. **1. Unified Representation:** STSUN effectively addresses the variability in data characteristics by leveraging inherent metadata from each dimension to achieve a unified representation. This capability allows STSUN to seamlessly adapt to input and output data with arbitrary image sizes, temporal lengths, and spectral bands, facilitating robust and scalable dense prediction across diverse data sources. **2. Unified Tasks:** STSUN unifies semantic segmentation, binary change detection, and semantic change detection within a single framework by incorporating a predefined, learnable task embedding set. By incorporating the selected task embedding and output temporal length into the output temporal dimension as metadata, the model is explicitly guided to perform the intended dense prediction task with the appropriate temporal configuration. **3. Flexible Category Set:** STSUN enables flexible prediction across multiple sets of semantic categories, accomplished by a predefined and trainable category embedding set. The selected category subset is integrated into the output channel dimension as metadata, which guides the model to identify the specific classes needed for a given prediction.

To further improve the model's ability to capture diverse spatial-temporal-spectral (STS) combinations, we design a Local-Global Window Attention (LGWA) mechanism. The module first extracts local features through three overlapping window-based attention layers with different shapes. It then applies a global attention layer to aggregate contextual information across the entire input.

This design facilitates the joint modeling of local and global representations, improving the model's effectiveness in complex remote sensing tasks.

The main contributions of this work are as follows:

1. We propose the STSUN, which accommodates arbitrary input and output configurations across spatial, temporal, and spectral dimensions, thereby addressing the prevalent issue of model rigidity in handling heterogeneous remote sensing data.

2. We introduce a unified task modeling approach using trainable task embeddings, enabling semantic segmentation, binary change detection, and semantic change detection within a single architecture, thereby eliminating the need for separate models or task-specific heads.

3. We design a flexible category prediction method, which utilizes trainable semantic embeddings to enable the model to perform dense prediction for multiple specified sets of output classes, removing the constraint of a fixed category schema and the associated costs of retraining.

## 2 RELATED WORKS

### 2.1 REMOTE SENSING DATA HETEROGENEITY

Remote sensing data exhibits significant heterogeneity across spatial, temporal, and spectral dimensions (Fassnacht et al., 2023). Spatially, differences in ground sampling distance and image extent hinder generalization (Yuan, 2018). Temporally, data availability ranges from bi-temporal pairs used for binary change detection to dense time series with hundreds of observations. This disparity in sequence length makes models trained on one temporal length incompatible with others. Spectrally, the number and type of bands vary from single panchromatic bands to hundreds of hyperspectral bands (Green et al., 1998), which makes models often tied to specific sensors. While some approaches attempt to unify inputs from different sensors using techniques like hypernetworks, they often overlook other dimensions (Xiong et al., 2024; Zhao et al., 2025; Li et al., 2025). The core research gap is thus a unified model architecture that can ingest and process this diverse spatial-temporal-spectral data at both the input and output.

### 2.2 DENSE PREDICTION TASK UNIFICATION

While dense prediction encompasses tasks including semantic segmentation, binary change detection, and semantic change detection (Shen et al., 2022), current unification efforts are limited. Multi-task learning frameworks have shown promise, leveraging shared representations to improve performance and efficiency (Wang et al., 2023). Recent methods in remote sensing have increasingly adopted multi-task learning to address multiple dense prediction tasks within a unified framework (Zhao et al., 2023a). However, these methods suffer from two major drawbacks. First, they exhibit a strong reliance on annotations of all tasks, which limits the pool of usable training data (Shen et al., 2022). Second, they often rely on task-specific decoders, which prevents the full exploitation of synergies between tasks during feature synthesis (Chen et al., 2022b). Consequently, a significant research gap exists for a unified model that can jointly learn multiple tasks from unpaired datasets.

### 2.3 FLEXIBLE SEMANTIC CLASS SETS

The semantic categories required for remote sensing tasks are highly dependent on specific applications and geographic scenes, ranging from simple binary sets to complex multi-class taxonomies. However, state-of-the-art models (Chen et al., 2022a; Zhao et al., 2023b; 2024a) are typically designed for fixed class sets, where the classification layer is hard-coded to a specific number of outputs. This rigidity limits their generalization across diverse tasks (Rebuffi et al., 2017). While Open-Vocabulary Segmentation (OVS) approaches address similar flexibility challenges, they predominantly rely on cross-modal alignment and require extensive image-text paired data. Distinct from OVS, our proposed framework achieves flexible semantic prediction through a visual mechanism. By utilizing learnable category embeddings and hypernetworks to dynamically adapt the classification head, we eliminate the dependency on external text encoders or linguistic descriptions, allowing for efficient adaptation within the visual domain.

## 3 METHODOLOGY

### 3.1 PROBLEM FORMULATION

Dense prediction in remote sensing encompasses a range of tasks that, despite their distinct objectives, share a common foundation: inferring structured, pixel-wise semantic information from multi-dimensional earth observation data. We formalize these tasks within a unified mathematical framework by conceptualizing them as a highly flexible tensor-to-tensor mapping problem.

Let an input remote sensing data instance be represented by a primary data tensor $X \in \mathbb{R}^{T_1 \times C_1 \times H_1 \times W_1}$, while the corresponding model output is a prediction tensor $Y \in \mathbb{R}^{T_2 \times C_2 \times H_2 \times W_2}$, consistent with the formulation presented in Section Introduction.

To address the profound variability in data-task-category specific requirements, we augment this core tensor representation with explicit metadata. The model's behavior is conditioned not only on the data tensor $X$ but also on two metadata sets: an input set $M_1$ describing the source data, and an output set $M_2$ specifying the target prediction structure. These sets are decomposed as follows:

- **Input Metadata:** $M_1 = \{M_{(H_1, W_1)}, M_{T_1}, M_{C_1}\}$. This set provides essential context about the input data $X$. $M_{(H_1, W_1)}$ encodes spatial information, including pixel locations and spatial resolution. $M_{T_1}$ contains the acquisition timestamps for each of the $T_1$ temporal slices. $M_{C_1}$ specifies the wavelengths for each of the $C_1$ spectral channels.
- **Output Metadata:** $M_2 = \{M_{(H_1, W_1)}, M_{T_2}, M_{C_2}\}$. This set specifies the desired output format and task. $M_{(H_1, W_1)}$ defines the pixel locations and spatial resolution of the prediction $Y$. $M_{T_2}$ determines the temporal nature of the task. It includes the target temporal length $T_2$ and a trainable task embedding that specifies which dense prediction task to perform. $M_{C_2}$ defines the prediction's semantic space, consisting of a dynamically selectable subset of trainable semantic class embeddings corresponding to the $C_2$ target classes.

Within this framework, our goal is to learn a single, unified mapping function $f_\theta$ parameterized by $\theta$, capable of adapting to arbitrary input and output configurations. The general mapping is defined as:

$$Y = f_\theta(X, M_1, M_2) \tag{3}$$

The three core tasks, including semantic segmentation, binary change detection, and semantic change detection, as well as various semantic category sets, are thus handled as specific instances of this general function. This metadata-driven formulation provides the mathematical foundation for a model that is not constrained by fixed data structures, predefined tasks, or category sets, enabling a truly unified approach to dense prediction in remote sensing.

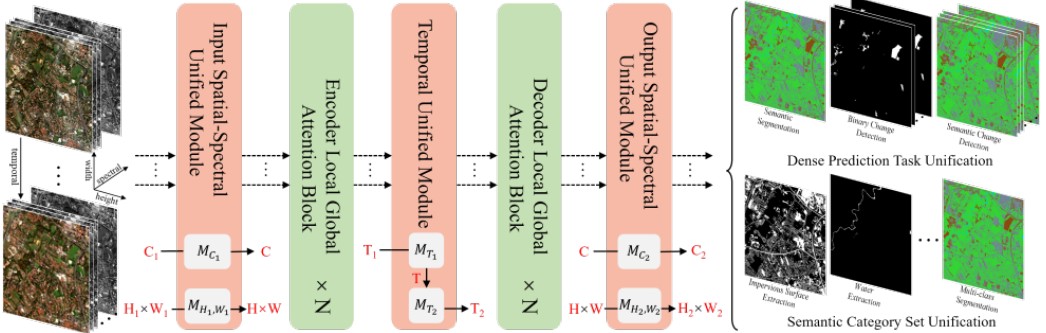

Figure 2: Overview of the STSUN architecture. The highlighted shapes on either side of each unified module indicate its unified dimension.

### 3.2 OVERVIEW OF THE SPATIAL-TEMPORAL-SPECTRAL UNIFIED NETWORK

STSUN achieves unified data representation across spatial, temporal, and spectral dimensions by leveraging input and output metadata. As illustrated in Figure 2, STSUN comprises five stages,

utilizing trainable task and class embeddings to adapt flexibly to diverse input/output configurations and class sets. Details can be found in Section A.1.

First, the Input Spatial-Spectral Unified Module (ISSUM) unifies the spatial and spectral dimensions. By exploiting inherent spectral continuity and spatial proximity, ISSUM employs hypernetworks conditioned on $M_{(H_1, W_1)}$ and $M_{C_1}$ to generate adaptive linear layers. These layers map the input into a unified representation while maintaining temporal independence to prevent multitemporal fusion degradation.

Second, a shared-weight encoder, composed of Encoder Local-Global Attention Blocks (Figure 4), processes each temporal image independently to extract robust local and global features.

Third, the Temporal Unified Module (TUM) bridges the encoder and decoder by unifying the temporal dimension. Leveraging temporal change continuity, TUM utilizes a hypernetwork conditioned on $M_{T_1}$ to generate adaptive parameters for feature-level fusion across temporal phases. Subsequently, a trainable task embedding guides the network toward the specific dense prediction task. Specifically, $M_{T_2}$ conditions a hypernetwork to adjust the output temporal length, mapping the unified representation to an output of arbitrary temporal length and facilitating the joint modeling of multiple tasks.

Fourth, a shared-weight decoder refines the fused, task-guided features using Decoder Local-Global Attention Blocks, suppressing irrelevant information and aligning representations with the task's semantic requirements.

Fifth, the Output Spatial-Spectral Unified Module (OSSUM) finalizes the spatial and spectral mapping. It restores features to the original resolution using an adaptive layer generated from $M_{(H_1, W_1)}$ and subsequently maps them to class-specific predictions using an $M_{C_2}$ conditioned layer. The $M_{C_2}$ defines the target category set, enabling flexible, class-agnostic prediction.

By structuring processing across these five stages, STSUN effectively accommodates diverse configurations, enabling high-performance joint modeling of multiple tasks and category sets.

## 3.3 DIMENSION UNIFIED MODULE

To facilitate the mapping from the original data space to the unified feature space across different dimensions (spatial, temporal, and spectral), and to transform the unified features into outputs of appropriate shapes according to the specific task and category requirements, STSUN introduces the Dimension Unified Module (DUM). This module serves as the fundamental implementation of any dimension unification in ISSUM, TUM, and OSSUM. DUM employs dimensional metadata and trainable embeddings to generate adaptive weights and biases for linear layers, thereby enabling adaptive and metadata-aware feature transformation.

Given an input tensor $\mathbf{X} \in \mathbb{R}^{L \times D_1}$ and its metadata $\mathbf{M} \in \mathbb{R}^{D_1}$, as well as trainable embeddings $\mathbf{E} \in \mathbb{R}^{D_1}$ when applied to output temporal and spectral dimensions, which indicates the task or semantic class set, DUM maps the varying input tensor $\mathbf{X}$ to a unified feature $\mathbf{Y} \in \mathbb{R}^{L \times D_2}$, or maps a unified feature $\mathbf{X}$ into the varying output $\mathbf{Y}$, as illustrated in Figure 3. $D_1$ and $D_2$ can refer to spatial, temporal, or spectral dimensions. The detailed procedure is as follows:

- **Metadata Embedding:** Metadata $\mathbf{M}$ is tokenized by a linear layer, and the positional encoding and optional trainable embeddings $\mathbf{E}$ are added to it. A learnable class token [CLS] is prepended to the token sequence $\mathbf{T} \in \mathbb{R}^{(D_1+1) \times D_2}$.

- **Cross-Variable Relationship Modeling:** The metadata token sequence $\mathbf{T}$ is processed through multiple transformer blocks to capture the underlying relationships among metadata.

- **Adaptive Linear Layer Generation:** $\mathbf{T}$ is divided into the [CLS] token and other tokens. The [CLS] token is linearly projected to generate bias parameters $\mathbf{b} \in \mathbb{R}^{D_2}$, while the remaining tokens are projected to form a weight matrix $\mathbf{W} \in \mathbb{R}^{D_1 \times D_2}$. The resulting $\mathbf{W}$ and $\mathbf{b}$ form a linear layer that maps the input features with $D_1$ channels to output features with $D_2$ channels.

- **Adaptive Feature Mapping:** The input $\mathbf{X} \in \mathbb{R}^{L \times D_1}$ can be processed through the generated linear layer, resulting in an output $\mathbf{Y} \in \mathbb{R}^{L \times D_2}$.

The effectiveness of DUM in unifying dimensions can be exemplified by the input spectral dimension. Multispectral and hyperspectral remote sensing data contain varying numbers of spectral channels, each associated with a specific wavelength. Due to the continuity of spectral wavelengths, these datasets can be interpreted as different sampling rates in the spectral domain. DUM leverages this continuity to effectively encode diverse spectral inputs within a unified representation. The unification of each dimension varies slightly, and their details are provided in the section A.1 of the supplementary material.

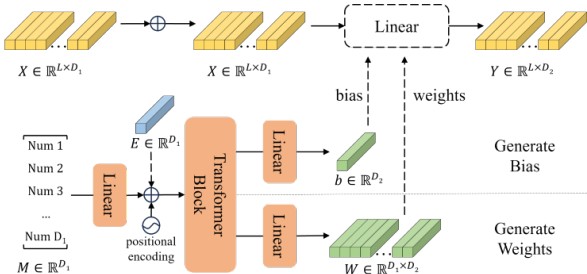

Figure 3: Illustration of the Dimension Unified Module.

By adaptively mapping STS variable subsets to a unified feature representation, DUM not only standardizes feature shapes but also preserves key relationships in the original remote sensing data. Additionally, DUM transforms this unified representation into outputs of task-specific shapes based on output dimensionality metadata. This enables STSUN to flexibly support inputs and outputs with diverse dimensional configurations, allowing it to generalize across various dense prediction tasks and class sets.

### 3.4 LOCAL-GLOBAL WINDOW ATTENTION

After unifying diverse spatial-temporal-spectral input and output data, STSUN requires robust feature extraction to effectively handle various remote sensing scenarios and tasks. Therefore, we propose the Local-Global Window Attention module, which performs self-attention within multiple local windows of varying shapes and a single global window, as illustrated in Figure 4. This design jointly captures fine-grained local features and coarse-grained global features, enhancing the model's performance across diverse remote sensing applications.

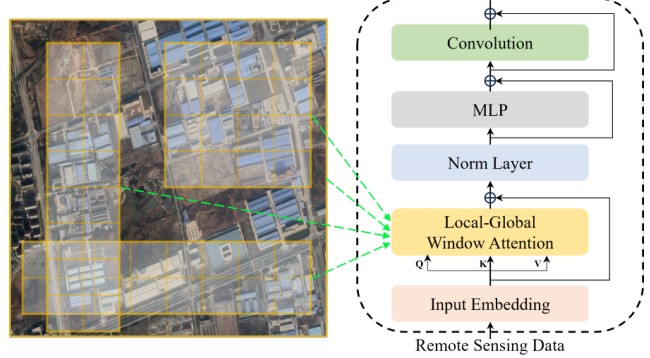

Figure 4: Architecture of the LGWA-based Local Global Attention Block.

The LGWA module integrates three local window attention mechanisms with varying sizes and configurations, alongside a single global attention mechanism. The distinct shapes of the local windows are designed to capture multi-level local features from ground objects comprehensively. Subsequently, the global attention mechanism models global contextual relationships among these objects based on the extracted local features, enhancing the overall model performance.

## 4 EXPERIMENTAL RESULTS

To validate STSUN's adaptability to arbitrary inputs and outputs configurations, as well as its capability to perform multiple dense prediction tasks with support for variable class subsets, we conducted experiments on seven datasets covering building and Land Use/Land Cover (LULC) scenarios, as summarized in Table 1. These datasets encompass a broad spectrum of configurations: spatial resolutions ranging from 0.075m to 10m, spectral coverage from 3 to 13 bands, and temporal sampling intervals spanning from 1 day to 2 years. For each scenario, a unified STSUN model was trained on the combined training sets of all datasets and evaluated on their respective test sets. Experimental details are provided in the section A.2 of the supplementary material.

Table 1: Brief Introduction of The Experimental Datasets. LULC denotes Land use/Land cover. SS, BCD, SCD denote semantic segmentation, binary change detection, semantic change detection, respectively.

| Name | Scenario | Task | $T_1$ | $T_2$ | $C_1$ | $C_2$ | $H, W$ | Resolution | Samples | Interval |
|---|---|---|---|---|---|---|---|---|---|---|
| WHU (Ji et al., 2019) | Building | SS | 1 | 1 | 3 | 2 | 512×512 | 0.3 | 8189 | N/A |
| WHU-CD (Ji et al., 2019) | Building | SCD | 2 | 2 | 3 | 2 | 1024×1024 | 0.075 | 480 | 4 year |
| LEVIR-CD (Chen & Shi, 2020) | Building | BCD | 2 | 1 | 3 | 2 | 1024×1024 | 0.5 | 445 | 5-14 year |
| TSCD (Zhao et al., 2024b) | Building | BCD | 4 | 3 | 3 | 2 | 256×256 | 0.5 | 2700 | 2 year |
| LoveDA (Wang et al., 2022a) | LULC | SS | 1 | 1 | 3 | 7 | 1024×1024 | 0.3 | 5987 | N/A |
| Dynamic. (Toker et al., 2022) | LULC | BCD &SCD | 24 | 24 | 4 | 6 | 1024×1024 | 3 | 54750 | 1 month |
| reBEN (Clasen et al., 2024) | LULC | SS | 1 | 1 | 13 | 19 | 120×120 | 10 | 549488 | N/A |

## 4.1 OVERALL COMPARISON ON THE BUILDING-RELATED SCENARIO

The efficacy of STSUN was evaluated through extensive experiments on four benchmark building datasets. It was compared against several state-of-the-art approaches, with quantitative results summarized in Tables 2, 3, 4, and 5, where the best and second-best scores are highlighted in bold and underlined, respectively. To differentiate the performance improvements brought by STSUN's architectural design and data-task unification, we also included $STSUN_{single}$, a model with an identical architecture trained exclusively on a single dataset.

Experimental results demonstrate that a single STSUN model can adapt to diverse input and output configurations and effectively handle multiple dense prediction tasks. It achieves state-of-the-art performance, outperforming competing models across a range of challenging remote sensing benchmarks. Notably, while the performance of $STSUN_{single}$ is comparable to the strongest baselines, the full STSUN model achieves superior results, surpassing all comparison methods. These findings indicate that STSUN's advantage stems from its unified modeling of data and tasks. The unification enables the model to leverage extensive data from diverse tasks, thereby facilitating synergistic improvements across tasks in a manner similar to multi-task learning.

Figure 5 presents inference results of STSUN from all four datasets. Visually, STSUN consistently produces accurate and complete segmentation and change maps. The results exhibit few false positives and false negatives, particularly in challenging areas such as those with small objects, intricate boundaries, or subtle temporal variations. For instance, in building segmentation on the WHU dataset (Figure 5(a)), STSUN generates sharp edges and complete building shapes.

Table 2: Comparison on the WHU Dataset.

| Methods | F1 (%)↑ | IoU (%)↑ |
|---|---|---|
| Deeplabv3+ (Chen et al., 2018) | 94.42 | 89.43 |
| MA-FCN (Wei et al., 2020) | 94.83 | 90.18 |
| Segformer (Xie et al., 2021) | 94.57 | 89.70 |
| TransUNet (Chen et al., 2021) | 93.56 | 87.89 |
| CMTFNet (Wu et al., 2023) | 92.59 | 86.21 |
| CaSaFormer (Li et al., 2024a) | 93.28 | 87.42 |
| $STSUN_{single}$ | 94.77 | 90.06 |
| STSUN | **95.29** | **91.00** |

Table 3: Comparison on the WHU-CD Dataset.

| Methods | F1 (%) | IoU (%) |
|---|---|---|
| Deeplabv3+ (Chen et al., 2018) | 90.07 | 81.93 |
| MA-FCN (Wei et al., 2020) | 87.97 | 78.52 |
| Segformer (Xie et al., 2021) | 89.68 | 81.30 |
| TransUNet (Chen et al., 2021) | 91.52 | 84.37 |
| CaSaFormer (Li et al., 2024a) | 91.42 | 84.20 |
| MRANet (Jiang et al., 2024) | 91.53 | 84.38 |
| $STSUN_{single}$ | 91.55 | 84.42 |
| STSUN | **92.13** | **85.40** |

Table 4: Comparison on LEVIR-CD Dataset.

| Methods | F1 (%) | IoU (%) |
|---|---|---|
| DSIFN (Zhang et al., 2020) | 88.13 | 78.77 |
| STANet (Chen & Shi, 2020) | 87.26 | 77.39 |
| SNUNet (Fang et al., 2022) | 88.16 | 78.83 |
| BIT (Chen et al., 2022a) | 89.30 | 80.68 |
| MTCNet (Wang et al., 2022b) | 90.24 | 82.22 |
| AMTNet-50 (Liu et al., 2023) | 90.76 | 83.08 |
| $STSUN_{single}$ | 91.25 | 83.92 |
| STSUN | **91.59** | **84.49** |

Table 5: Comparison on the TSCD Dataset.

| Methods | F1 (%) | IoU (%) |
|---|---|---|
| SNUNet-CD (Fang et al., 2022) | 63.22 | 47.22 |
| USSFCNet (Lei et al., 2023) | 55.68 | 39.80 |
| A2Net (Li et al., 2023b) | 53.16 | 37.19 |
| SEIFNet (Huang et al., 2024) | 60.37 | 44.01 |
| Contrast. (Zhao et al., 2024b) | 64.35 | 48.45 |
| TS-COUD (Zhao et al., 2024b) | 65.24 | 49.33 |
| $STSUN_{single}$ | 65.98 | 49.23 |
| STSUN | **66.48** | **49.79** |

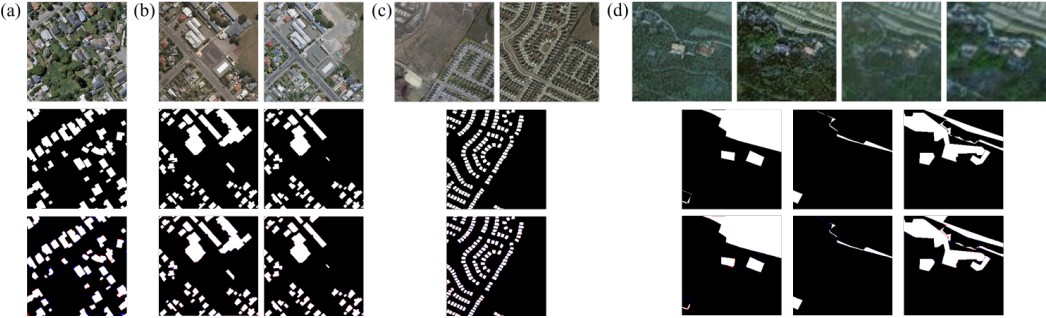

Figure 5: Sample inference results on for building scene datasets. The input images, ground truths and predictions are shown in the first, second and third rows, respectively. Red areas denote false positives and blue areas denote false negatives. (a) WHU dataset sample. (b) WHU-CD dataset sample. (c) LEIVR-CD dataset sample. (d) TSCD dataset sample.

## 4.2 OVERALL COMPARISON ON THE LULC SCENARIO

The efficacy of the proposed STSUN is further validated through comparisons with state-of-the-art methods on three challenging LULC datasets. Quantitative results are presented in Tables 6, 7 and 8, with the best and second-best performances highlighted in bold and underlined, respectively. Inference results are provided in the section A.4.1 of the supplementary material.

The results shows that STSUN demonstrates strong adaptability in LULC scenarios by accommodating diverse input and output configurations to perform multiple dense prediction tasks across varying category sets. STSUN demonstrates state-of-the-art performance, surpassing all baseline methods in LULC classification across the experimental datasets. Its success is attributed to a unified modeling framework that holistically considers data, tasks, and

Table 6: Comparison on the reBEN Dataset.

| Methods | AP(%)↑ | AF(%)↑ |
|---|---|---|
| ResNet (He et al., 2016) | 72.75 | 64.39 |
| ViT (Dosovitskiy et al., 2020) | 68.29 | 57.46 |
| MLP-Mixer (Tolstikhin et al., 2021) | 71.50 | 61.91 |
| ConvMixer (Trockman & Kolter, 2022) | 69.66 | 62.31 |
| MobileViT (Mehta & Rastegari, 2021) | 75.89 | 62.28 |
| ConvNext-V2 (Woo et al., 2023) | 69.05 | 62.90 |
| STSUN | **75.96** | **65.78** |

classes. This framework facilitates the integration of diverse datasets, exploits inter-task complementarity for joint performance enhancement, and mines intrinsic class correlations to improve predictive accuracy for individual categories.

Table 7: Comparison on the LoveDA Dataset.

| Methods | AF(%)↑ | mIoU(%)↑ |
|---|---|---|
| DeepLabV3+ (Chen et al., 2018) | 53.50 | 47.62 |
| ResUNet-a (Diakogiannis et al., 2020) | 62.12 | 54.16 |
| A2FPN (Li et al., 2022) | 70.03 | 61.14 |
| MSAFNet (Lyu et al., 2023) | 70.40 | 60.76 |
| CLCFormer (Long et al., 2023) | 72.16 | 63.85 |
| SSCBNet (Li et al., 2024b) | 74.05 | 64.58 |
| STSUN | **75.14** | **66.21** |

Table 8: Comparison on the Dynamic. Dataset.

| Methods | SCS↑ | mIoU(%)↑ |
|---|---|---|
| CAC (Lai et al., 2021) | 17.7 | 37.9 |
| TSViT (Tarasiou et al., 2023) | 23.0 | 50.5 |
| UTAE (Garnot & Landrieu, 2021) | 25.9 | 53.7 |
| A2Net (Li et al., 2023a) | 22.2 | 47.2 |
| SCanNet (Ding et al., 2024) | 24.8 | 53.0 |
| TSSCD (He et al., 2024) | 12.0 | 33.9 |
| STSUN | **31.4** | **56.3** |

## 4.3 EFFECT OF UNIFYING DENSE PREDICTION TASKS

To validate the effect of unifying dense prediction tasks, we conducted a comparison experiment on four datasets in the building scenario. Specifically, we compare the performance of models trained individually on each dataset against a single STSUN model trained jointly on all datasets. the former is denoted as $STSUN\_single$ and the latter as $STSUN_{\_unified}$.

The results presented in Table 9 demonstrate that $STSUN_{\_unified}$ consistently outperforms its single-task counterparts across each dataset. This performance gain is attributed to two factors. First, joint training on data from multiple dense prediction tasks exposes the model to a richer and more diverse set of samples, enhancing its ability to learn comprehensive spatial-temporal-spectral

features from remote sensing imagery. Second, the model capitalizes on the complementary nature of these tasks. This synergy, analogous to a multi-task learning paradigm, enables the learning of more powerful and robust feature representations, thereby elevating the model's performance across all individual tasks.

## 4.4 EFFECT OF FLEXIBLE SEMANTIC CATEGORY SET

To demonstrate the effect of handling multiple semantic category sets , we designed an experiment on the LoveDA and DynamicEarthNet datasets in the LULC scenario. We compare three model configurations: 1) $STSUN_{single}$, trained exclusively on a single dataset to predict its own category set. 2) $STSUN_{fixed}$, trained jointly on both datasets to predict a fixed, 10-category set corresponding to the union of their category sets. 3) $STSUN_{flexible}$, trained jointly on both datasets but predicted flexibly with the 10-category embedding set, each embedding indicates a predicted category.

Table 9: Comparison over Task Unification on Four Building Datasets.

| Dataset | Methods | F1 (%)↑ | IoU (%)↑ |
|---|---|---|---|
| WHU | $STSUN_{single}$ | 94.77 | 90.06 |
| | $STSUN_{unified}$ | **95.29** | **91.00** |
| WHU-CD | $STSUN_{single}$ | 91.55 | 84.42 |
| | $STSUN_{unified}$ | **92.13** | **85.40** |
| LEVIR-CD | $STSUN_{single}$ | 91.25 | 83.92 |
| | $STSUN_{unified}$ | **91.59** | **84.49** |
| TSCD | $STSUN_{single}$ | 65.98 | 49.23 |
| | $STSUN_{unified}$ | **66.48** | **49.79** |

As shown in Table 10, $STSUN_{flexible}$ outperforms $STSUN_{fixed}$ on both datasets, while achieving performance comparable to the models $STSUN_{single}$. The degraded performance of $STSUN_{fixed}$ is expected. This model is constrained to predict over the union of all categories, even for an image from a scene that does not contain certain categories. Forcing a single, fixed-size output space across scenes with disparate semantic sets introduces ambiguity and negatively impacts performance on each respective scene. In contrast, $STSUN_{flexible}$ dynamically adapts its predictive output to the relevant subset of semantic categories for each scene, which explains why its performance is on par with the individually trained models. Crucially, however, $STSUN_{flexible}$ offers superior model efficiency, as a single, unified model can be deployed across scenes with different semantic category sets without requiring any additional training or fine-tuning.

## 4.5 ABLATION STUDY

To evaluate the effectiveness of STSUN's dimension unification strategy and the LGWA module, we conducted ablation studies on the TSCD dataset. The proposed strategy, referred to as Decoupled Unification (DU), maintains the independence of the temporal dimension. For comparison, we introduced a baseline termed Coupled Unification (CU), which directly unifies the spatial, temporal, and spectral dimensions at both input and output stages. Additionally, LGWA was replaced with standard global attention or window attention (Liu et al., 2021) to assess its contribution.

Table 10: Comparison over Category Unification on Two LULC Datasets.

| Model | LoveDA Dataset | | Dynamic. Dataset | |
|---|---|---|---|---|
| | AF↑ | mIoU↑ | SCS↑ | mIoU↑ |
| $STSUN_{single}$ | 74.66 | 65.59 | 30.0 | 55.3 |
| $STSUN_{fixed}$ | 74.03 | 64.67 | 29.4 | 54.2 |
| $STSUN_{flexible}$ | **74.81** | **65.73** | **30.3** | **55.7** |

Table 11 displays the results of our ablation studies, demonstrating that both the decoupled unification strategy and LGWA outperform their respective baseline alternatives. Furthermore, their combined use leads to even greater performance improvements. LGWA surpasses standard global self-attention and window attention by utilizing local attention with multiple window size alongside global attention.

Moreover, the effectiveness of the Decoupled Unification Strategy stems from its preservation of independent temporal dimensions. The Coupled Unification Strategy unifies the input's temporal dimension at the data input stage. This is equivalent to pixel-level fusion

Table 11: Ablation study on the TSCD Dataset.

| Strategy | Attention | F1(%)↑ | IoU(%)↑ |
|---|---|---|---|
| CU | Global | 62.87 | 45.85 |
| CU | Window | 62.63 | 45.42 |
| CU | LGWA | 63.22 | 46.22 |
| DU | Global | 65.91 | 49.15 |
| DU | Window | 65.68 | 48.79 |
| DU | LGWA | **66.48** | **49.79** |

of multi-temporal images, causing the fused features to be contaminated by background-irrelevant information from each temporal phase (Chen et al., 2022b; Zhao et al., 2023b). Furthermore, this strategy only maps the unified temporal length back to a variable temporal length at the model's final output layer to generate task-specific dense predictions. This late-stage mapping limits the model's ability to utilize sufficient parameters to adapt to diverse tasks. Conversely, by maintaining independent temporal dimensions, the Decoupled Unification Strategy allows the model to unify the input's temporal dimension after extracting features from each temporal phase. This enables multi-temporal feature-level fusion, significantly mitigating the interference from irrelevant information (Chen et al., 2022b; Zhao et al., 2023b). Subsequently, the fused features are mapped into a sequence of tokens with the desired output temporal length, which is then processed by a decoder. This comprehensive decoding process provides the model with ample parameters to effectively adapt to a variety of downstream tasks.

## 5 CONCLUSION

We propose STSUN, a novel architecture designed to address the rigidity of existing models in remote sensing dense prediction. By leveraging metadata-informed unified representation generated by hypernetworks, STSUN unifies the spatial, temporal, and spectral dimensions, enabling flexible handling of variable input–output configurations. Through embedding-based mechanisms, it integrates semantic segmentation, binary change detection, and semantic change detection within a single framework, while supporting multiple sets of prediction classes without retraining. Extensive experiments on diverse datasets demonstrate that a single STSUN model can consistently adapt to heterogeneous inputs, outputs, and tasks, achieving competitive performance. STSUN provides a robust baseline for universal dense prediction in remote sensing, reducing reliance on data-, task-, and category-specific model designs and retraining efforts.

## 6 STATEMENT

### 6.1 ETHICS STATEMENT

The development and training phases of the presented models necessitate substantial computational resources, inherently leading to significant energy consumption. This energy expenditure constitutes a critical environmental concern, contributing materially to the carbon footprint and other associated ecological impacts. Recognizing these externalities, we emphasize the importance of mitigating strategies focused on energy sourcing. Specifically, transitioning towards renewable and low-carbon energy infrastructure for powering computational tasks is paramount to lessening the environmental burden associated with large-scale model training.

Adopting sustainable energy solutions can demonstrably reduce the ecological ramifications of the computational pipeline, aligning technological advancement with environmental stewardship. It is incumbent upon the research community to proactively evaluate and address the environmental costs inherent in deploying computationally demanding methodologies. Key mitigation approaches encompass not only the adoption of sustainable energy but also advancements in energy-efficient hardware architectures and the continuous pursuit of algorithmic optimization to reduce computational overhead. Promoting such holistic sustainable computational practices is crucial for ensuring that progress in artificial intelligence does not inadvertently exacerbate environmental challenges, but rather contributes responsibly to future development.

### 6.2 REPRODUCIBILITY STATEMENT

To facilitate reproducibility, we provide comprehensive experimental details in the section A.2 of supplementary materials. The code is provided at the supplementary code.

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

# A APPENDIX

## A.1 DETAILS OF DIMENSION UNIFIED MODULE

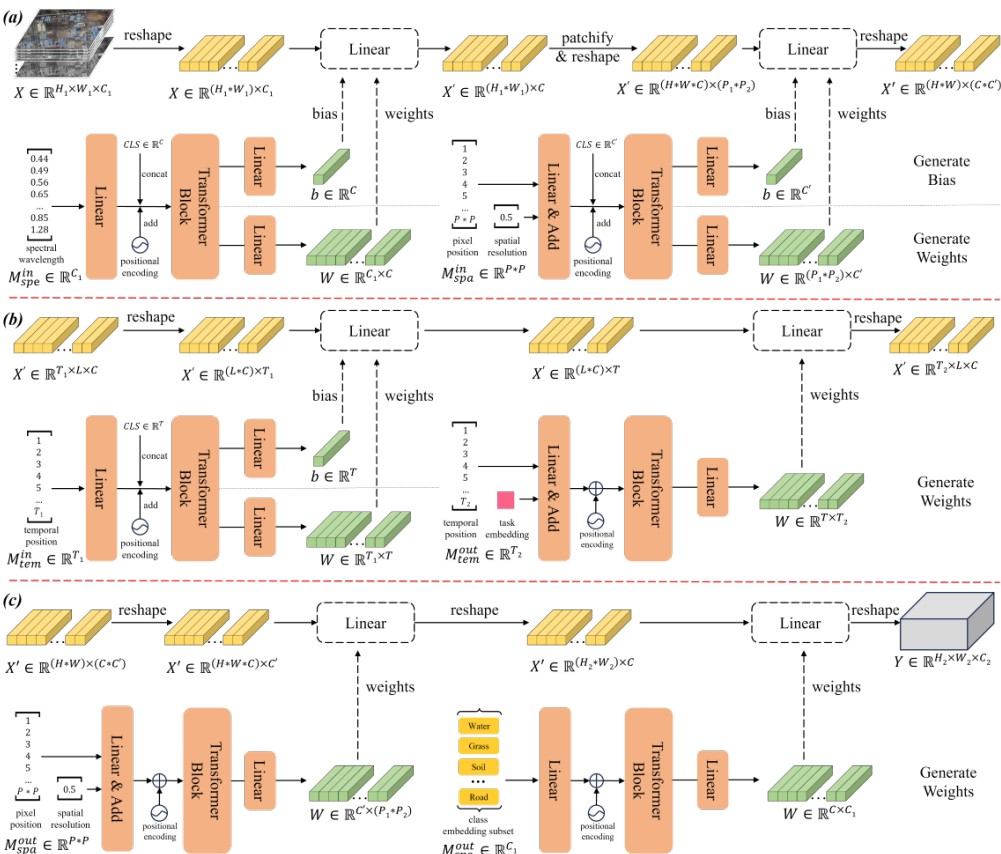

Figure 6: Illustration of the Spatial-Temporal-Spectral Unified Module. (a) Input Spatial-Spectral Unified Module. (b) Temporal Unified Module. (c) Output Spatial-Spectral Unified Module.

To facilitate the mapping from the raw data space to a unified feature space across different dimensions (spatial, temporal, and spectral), and to transform these unified features into outputs of appropriate shapes according to specific task and class requirements, our proposed STSUN employs the Spatial-Temporal-Spectral Unified Module (STSUM) to handle STS dimension, which consists of ISSUM, TUM and OSSUM . First, the ISSUM maps the variable input image size $(H_1, W_1)$ and spectral bands $C_1$ to a predefined, unified size $(H, W)$ and spectral bands $C$ in the spatial and spectral dimensions, respectively. Subsequently, the TUM maps the variable input temporal length $T_1$ to a predefined, unified length $T$, which is then mapped to a variable output temporal length $T_2$ based on task demands. Finally, the OSSUM maps the unified image size $(H, W)$ and spectral bands $C$ to a variable output size $(H_2, W_2)$ and spectral bands $C_2$, guided by the prediction class set and image size requirements. The details of these modules are illustrated in Figure 6 (a), (b), and (c).

The core mechanism of STSUM involves using metadata from each dimension, along with optional trainable embeddings, to generate adaptive linear layers. This enables the transformation of variable input data into unified features or, conversely, the conversion of unified features into variable outputs tailored to specific requirements. The feature mapping mechanism is consistent across the modules and comprises two main branches: a hyper-network branch and a mapping network branch. In the hyper-network branch, metadata from each dimension is first tokenized using a linear layer and augmented with positional encodings. These tokens are then processed through several Transformer blocks to capture the latent relationships among them. Finally, another linear layer generates the parameters for the adaptive mapping network. In the mapping network branch, this dynamically

generated network applies a linear transformation to the input features, thereby unifying the input data or generating the adaptive output. Although the underlying principle is similar, each module requires distinct and meticulous design considerations regarding feature shapes and parameter generation. This ensures that the modules can achieve generic and robust dimensional unification by processing different metadata for different mapping requirements across various dimensions.

### A.1.1 Input Spatial-Spectral Unified Module

The ISSUM unifies the spectral and spatial dimensions of the input data. It first operates on the spectral dimension, mapping the variable input spectral bands $C_1$ to a predefined, unified spectral bands $C$ using spectral metadata. It then proceeds to the spatial dimension, mapping the variable input image size $(H_1, W_1)$ to a predefined, unified size $(H, W)$ using spatial metadata, as depicted in Figure 6(a). For a batch of input data $X \in \mathbb{R}^{T_1 \times H_1 \times W_1 \times C_1}$, since the ISSUM is designed to unify the spatial and spectral dimensions, the temporal dimension $T_1$ can be fused with the batch dimension. Consequently, the ISSUM only needs to process a single time-step input $X \in \mathbb{R}^{H_1 \times W_1 \times C_1}$. For the spectral dimension of $X$, the hyper-network branch utilizes the input data's spectral wavelengths as metadata $M_{\text{spe}}^{\text{in}} \in \mathbb{R}^{C_1}$. This metadata is first tokenized to $M' \in \mathbb{R}^{C_1 \times C_e}$ by a linear layer, where $C_e$ is a predefined unified spectral bands. Subsequently, a learnable class token, $\text{CLS} \in \mathbb{R}^{C_e}$, is concatenated to the token sequence, and positional encodings are added to incorporate relative position information, resulting in $M' \in \mathbb{R}^{(C_1+1) \times C_e}$. $M'$ is then processed by multiple Transformer blocks to capture latent relationships among the different spectral bands. The output is split into two parts: the CLS token and the remaining token sequence $M''$. The CLS token is passed through a linear layer to generate the bias parameter $b \in \mathbb{R}^{C_e}$, while $M''$ is passed through another linear layer to generate the weight matrix $W \in \mathbb{R}^{C_1 \times C_e}$. The generated $W$ and $b$ constitute a linear layer capable of mapping input features with $C_1$ channels to output features with $C_e$ channels. Accordingly, in the mapping network branch, the ISSUM reshapes $X \in \mathbb{R}^{H_1 \times W_1 \times C_1}$ to $X \in \mathbb{R}^{(H_1 \cdot W_1) \times C_1}$ and then transforms it using the generated mapping network to obtain $X' \in \mathbb{R}^{(H_1 \cdot W_1) \times C_e}$, thus achieving unification in the channel dimension.

Next, for the spatial dimension of $X'$, the ISSUM begins in the mapping network branch by applying 'patchify' and 'reshape' operations to $X' \in \mathbb{R}^{(H_1 \cdot W_1) \times C_e}$, transforming it into $X' \in \mathbb{R}^{(H \cdot W \cdot C_e) \times (P_h \cdot P_w)}$, where $P_h = H_1/H$ and $P_w = W_1/W$. This procedure converts the variable image size $(H_1, W_1)$ into a unified spatial size $(H, W)$ by moving the variable part into the channel dimension, which allows the ISSUM to unify a variable spectral bands of $P_h \cdot P_w$ in a similar way. Consequently, in the hyper-network branch, the ISSUM uses the patch positions and spatial resolution of the input data as spatial metadata, $M_{\text{spa}}^{\text{in}} \in \mathbb{R}^{P_h \cdot P_w}$. These are tokenized by separate linear layers and then summed to form the metadata token sequence, where the patch position denotes the location information of each pixel within a single patch. Finally, employing the same procedure as described for the spectral dimension, the ISSUM uses this spatial metadata token sequence to generate a bias parameter $b \in \mathbb{R}^{C_a}$ and a weight matrix $W \in \mathbb{R}^{(P_h \cdot P_w) \times C_a}$, where the $C_a$ is predefined unified spectral bands. These parameters define a linear layer that maps $X' \in \mathbb{R}^{(H \cdot W \cdot C_e) \times (P_h \cdot P_w)}$ to $X' \in \mathbb{R}^{(H \cdot W \cdot C_e) \times C_a}$. After reshaping, this yields $X' \in \mathbb{R}^{(H \cdot W) \times (C_e \cdot C_a)}$, thereby completing the unification of the input data in the spatial dimension.

### A.1.2 Temporal Unified Module

The TUM unifies the temporal dimension of the input and output data. It employs a similar mechanism with ISSUM to first map the variable input temporal length $T_1$ to a predefined, unified length $T$, and then map this unified length to a variable output length $T_2$ based on task requirements, as shown in Figure 6(b). For simplicity, let $C = C_e \cdot C_a$, $L = H \cdot W$. The TUM isolates the temporal dimension, obtaining a single data sample $X' \in \mathbb{R}^{T_1 \times L \times C}$, and reshapes it to $X' \in \mathbb{R}^{(L \cdot C) \times T_1}$, thereby transposing the temporal dimension to the position of the channel dimension. This allows the TUM to unify the temporal dimension of the input data using a similar method of ISSUM. TUM utilizes the temporal position information of the input data as metadata, $M_{\text{tem}}^{\text{in}} \in \mathbb{R}^{T_1}$, to adaptively generate a linear layer composed of a bias parameter $b \in \mathbb{R}^T$ and a weight matrix $W \in \mathbb{R}^{T_1 \times T}$. This mapping network transforms the temporally variable input $X' \in \mathbb{R}^{(L \cdot C) \times T_1}$ into a unified representation $X' \in \mathbb{R}^{(L \cdot C) \times T}$.

Subsequently, according to the requirements of the specific dense prediction task, the TUM needs to map the unified temporal length $T$ to a variable length $T_2$. Unlike the previous operation of mapping variable features to unified features, this process is reversed, which necessitates differences in the metadata and the generated parameters. Specifically, the TUM uses the output temporal length information and a selected task embedding as the output temporal metadata, $M_{\text{tem}}^{\text{out}} \in \mathbb{R}^{T_2}$. These are mapped through linear layers and then summed to form the metadata token sequence. The task embedding is a predefined, trainable embedding selected from a set, such as {semantic segmentation embedding, binary change detection embedding, semantic change detection embedding}, to specify the dense prediction task being performed. The metadata token sequence is then processed by a Transformer block to capture latent relationships between tokens. Following this, it passes through a linear layer to generate only the weight parameter $W \in \mathbb{R}^{T_2 \times T}$, which is reshaped to $W \in \mathbb{R}^{T \times T_2}$ to serve as the weights of the mapping network. Finally, this bias-free mapping network transforms $X' \in \mathbb{R}^{(L \cdot C) \times T}$ into $X' \in \mathbb{R}^{(L \cdot C) \times T_2}$, which is then reshaped back to $X' \in \mathbb{R}^{T_2 \times L \times C}$, thus converting the unified temporal length $T$ into a variable length $T_2$ according to specific task demands. The bias parameter is not generated because the mapping network would require a variable bias $b \in \mathbb{R}^{T_2}$, which cannot be generated from a fixed CLS token through a fixed linear layer.

### A.1.3 OUTPUT SPATIAL-SPECTRAL UNIFIED MODULE

The OSSUM unifies the spatial and spectral dimensions of the output data. It first maps the unified image size $(H, W)$ to a variable size $(H_2, W_2)$ using spatial metadata, and then maps the unified spectral spectral bands $C$ to a variable count $C_2$ using spectral metadata, as illustrated in Figure 6(c). The dimensional mapping mechanism of the OSSUM is similar to that of the TUM for the output temporal dimension, with the main difference lying in the metadata. Specifically, for the spatial dimension, the hyper-network branch of the OSSUM uses the patch position and spatial resolution as metadata, $M_{\text{spa}}^{\text{out}} \in \mathbb{R}^{P_h \cdot P_w}$. After passing through linear layers and Transformer blocks, it generates a weight parameter $W \in \mathbb{R}^{C_a \times (P_h \cdot P_w)}$ for the mapping network. Since the output spatial size is identical to the input spatial size, we have $M_{\text{spa}}^{\text{out}} = M_{\text{spa}}^{\text{in}}$. In the mapping network branch, the input feature $X' \in \mathbb{R}^{(H \cdot W) \times (C_e \cdot C_a)}$ is reshaped to $X' \in \mathbb{R}^{(H \cdot W \cdot C_e) \times C_a}$. It is then transformed by the mapping network to yield $X' \in \mathbb{R}^{(H \cdot W \cdot C_e) \times (P_h \cdot P_w)}$. An 'unpatchify' operation performs up-sampling, and a final reshape operation converts it to $X' \in \mathbb{R}^{(H_2 \cdot W_2) \times C_e}$, producing a variable-sized output in the spatial dimension.

Similarly, in the spectral dimension, the OSSUM first selects a subset from a predefined set of semantic class embeddings based on the requirements of the prediction class set. This subset of embeddings indicates all the classes the model needs to predict. For instance, while multiple remote sensing scenes might require dense prediction for various land cover types, leading to a total semantic class embedding set like {Tree, Water, Soil, Road, Building, Background}, a specific building extraction task would only require selecting the subset {Building, Background}. This subset directs the model to classify land cover into these two categories, enabling effective building extraction. Therefore, the selected subset of semantic class embeddings serves as the output spectral metadata, $M_{\text{spe}}^{\text{out}} \in \mathbb{R}^{C_2 \times C_e}$. Through a similar mechanism above, a weight parameter $W \in \mathbb{R}^{C_e \times C_2}$ is generated for the mapping network. In the mapping network branch, $X' \in \mathbb{R}^{(H_2 \cdot W_2) \times C_e}$ undergoes a linear transformation by the mapping network to produce $X' \in \mathbb{R}^{(H_2 \cdot W_2) \times C_2}$. Finally, a reshape operation yields the output $Y \in \mathbb{R}^{H_2 \times W_2 \times C_2}$, producing a variable output in the spectral dimension.

## A.2 EXPERIMENTAL DETAILS

### A.2.1 BUILDING SCENARIO

Dataset

The WHU Building dataset (Ji et al., 2019) is divided into two main components: one containing satellite imagery and another composed of aerial photos. In our study, we utilize the aerial photo subset, which consists of 8,189 images. These images are split into 4,736 for training, 1,036 for validation, and 2,416 for testing, each having a spatial resolution of 0.3 meters. In total, this subset represents over 22,000 buildings covering an area in excess of 450 square kilometers. Our experi-

ments were conducted using the original partitioning scheme and image dimensions (512×512) as specified by the WHU dataset.

The WHU-CD dataset (Ji et al., 2019) includes bitemporal very high-resolution (VHR) aerial images taken in 2012 and 2016, which clearly highlights major changes in building structures. The dataset is partitioned into non-overlapping patches of 1024×1024 pixels. These patches are further allocated into training, validation, and test sets following a 7:1:2 ratio.

The LEVIR-CD dataset (Chen & Shi, 2020) is an extensive resource for change detection, comprising VHR Google Earth images with a resolution of 0.5 m/pixel. These images capture a variety of building transformations over periods ranging from 5 to 14 years, with a particular emphasis on construction and demolition events. The bitemporal images have been expertly annotated using binary masks, where a label of 1 denotes a change and 0 signifies no change. In total, there are 31,333 labeled instances of building modifications. Our experimental setup used the dataset's original image dimensions of 1024×1024 and adhered to the provided data partitioning scheme.

The TSCD dataset (Zhao et al., 2024b) is constructed from WorldView-2 satellite imagery with a spatial resolution of approximately 0.5 m/pixel, acquired in 2016, 2018, 2020, and 2022. To mitigate external influences, the images underwent co-registration using manually selected control points and resampling to ensure a consistent coordinate framework. Building footprints were densely labeled for each temporal phase. Subsequently, three sets of change labels (2016–2018, 2018–2020, 2020–2022) were generated by performing differential operations on adjacent building distribution maps. The final TSCD dataset was created through uniform cropping and partitioning of these original images and derived labels.

Baseline

Since STSUN is adaptable to multiple datasets, it was trained across all datasets in each scenario, whereas the baselines were trained on individual datasets.

On the four building scene datasets, the compared CNN-based models include MA-FCN (Wei et al., 2020), Deeplabv3+ (Chen et al., 2018), STANet (Chen & Shi, 2020), SNUNet (Fang et al., 2022), DSIFN (Zhang et al., 2020), USSFCNet (Lei et al., 2023), MRANet (Jiang et al., 2024) and SEIFNet (Huang et al., 2024), the compared transformer-based models include Segformer (Xie et al., 2021), MTCNet (Wang et al., 2022b) and A2Net (Li et al., 2023b), and the CNN-transformer hybrid models include TransUNet (Chen et al., 2021), CMTFNet (Wu et al., 2023), CaSaFormer (Li et al., 2024a), BIT (Chen et al., 2022a), AMTNet-50 (Liu et al., 2023), Contrast-COUD (Zhao et al., 2024b) and TS-COUD (Zhao et al., 2024b).

### A.2.2 LULC Scenario

Dataset

The LoveDA dataset (Wang et al., 2022a) consists of 5,987 high-resolution optical remote sensing images (with a ground sampling distance of 0.3 m) each sized at 1024×1024 pixels. It covers seven land cover classes: building, road, water, barren, forest, agriculture, and background. The dataset is divided into 2,522 training images, 1,669 images for validation, and 1,796 images for testing, all drawn from two distinct scenes—urban and rural—from three Chinese cities: Nanjing, Changzhou, and Wuhan. The dataset poses considerable challenges due to the presence of multiscale objects, complex backgrounds, and uneven class distribution.

The DynamicEarthnet dataset (Toker et al., 2022) comprises 55 daily Sentinel-2 Image Time Series (SITS) collected globally between January 1, 2018, and December 31, 2019. For each month, data from the first day is annotated, which results in 24 ground truth segmentation maps per Area of Interest (AoI). Each image is 1024×1024 pixels and multi-spectral, containing four channels (RGB plus near-infrared). The annotations cover general land-use and land-cover categories: impervious surface, agriculture, forest, wetlands, soil, and water. The 'snow' class appears in only a few AoIs and has been excluded from this study.

The reBEN dataset (Clasen et al., 2024) is a large-scale, multi-modal remote sensing dataset comprising 549,488 pairs of Sentinel-1 and Sentinel-2 image patches. Each patch covers an area of 1200 m x 1200 m and is annotated with both pixel-level reference maps and scene-level multi-labels de-

rived from the 2018 CORINE Land Cover (CLC) map, using a 19-class nomenclature. To enhance the quality over its predecessor, BigEarthNet, the Sentinel-2 imagery underwent atmospheric correction using an updated sen2cor tool, and label noise was reduced by using the most recent CLC map.

Baseline

Since STSUN is adaptable to multiple datasets, it was trained across all datasets in each scenario, whereas the baselines were trained on individual datasets.

On the two LULC scene datasets, the compared CNN-based models include ResNet (He et al., 2016), Deepabv3+ (Chen et al., 2018), ResUNet-a (Diakogiannis et al., 2020), A2FPN (Li et al., 2022), MSAFNet (Lyu et al., 2023) and CAC (Lai et al., 2021), the compared transformer-based models include ViT (Dosovitskiy et al., 2020), MLP-Mixer (Tolstikhin et al., 2021), CLCFormer (Long et al., 2023) and TSViT (Tarasiou et al., 2023), the CNN-transformer hybrid models include ConvMixer (Trockman & Kolter, 2022), MobileViT (Mehta & Rastegari, 2021), ConvNext-V2 (Woo et al., 2023), SSCBNet (Li et al., 2024b), UTAE (Garnot & Landrieu, 2021), A2Net (Li et al., 2023a), SCanNet (Ding et al., 2024) and TSSCD (He et al., 2024).

## A.3 Implementation Details

### A.3.1 Data Augmentation

To validate the proposed methods, we adopted a minimalistic yet effective data augmentation strategy, deliberately refraining from complex augmentation schemes. Specifically, the employed transformations were limited to horizontal/vertical flipping (probability = 0.5) and transposition (probability = 0.5).

### A.3.2 Training and Inference

The STSUN model was implemented using PyTorch (Paszke et al., 2019) and executed on a single RTX A100 GPU (80G). Due to the heterogeneous image resolutions across the datasets, the batch size was set to 16 for the four building scene datasets and 4 for the two LULC scene datasets. Our optimization strategy combined binary cross-entropy loss with Dice coefficient loss, facilitating a balanced performance optimization.

The AdamW optimizer (Kingma & Ba, 2017) was initialized with a learning rate of 0.0001 and a weight decay of 0.001. A learning rate scheduler was employed to reduce the learning rate by a factor of 0.1 if no increase in the mean F1-score was observed on the aggregate validation set for 5 consecutive epochs. The training process spanned 100 epochs, ensuring robust convergence, and the best performing checkpoints—corresponding to the maximum mean F1-scores achieved—were retained for the testing phase. Furthermore, to ensure comparability with existing methodologies, all models were initialized using the default PyTorch settings across all datasets.

### A.3.3 Evaluation Metrics

To comprehensively evaluate the performance of the proposed models, we employ a set of standard metrics alongside dataset-specific protocols. Standard quantitative assessments include Overall Accuracy (OA), Precision (P), Recall (R), F1-score, and Intersection over Union (IoU). For multi-temporal and multi-category tasks, we report the Average F1-score (AF) and mean IoU (mIoU).

Furthermore, following the evaluation protocol of the DynamicEarthnet dataset (Toker et al., 2022), we adopt the Semantic Change Segmentation (SCS) metric to simultaneously assess the quality of binary change detection and semantic classification. The SCS metric addresses two distinct error types: failing to detect binary changes and predicting incorrect semantic classes for changed pixels. It is composed of two sub-metrics: Binary Change (BC) score and Semantic Change (SC) score.

**Binary Change (BC).** The BC score measures the quality of the predicted binary change map $\hat{b}$ by comparing its overlap with the ground-truth change map $b$. It is defined as the IoU of the binary

change masks:

$$BC(\mathbf{b}, \hat{\mathbf{b}}) = \frac{|\{\mathbf{b} = 1\} \cap \{\hat{\mathbf{b}} = 1\}|}{|\{\mathbf{b} = 1\} \cup \{\hat{\mathbf{b}} = 1\}|}, \tag{4}$$

where $\{\mathbf{b} = 1\}$ denotes the set of pixel indices where binary change occurs.

**Semantic Change (SC).** The SC score evaluates the semantic segmentation accuracy specifically on the regions where change occurred. It computes the IoU between the ground-truth labels $\mathbf{y}$ and predicted labels $\hat{\mathbf{y}}$, conditioned on the ground-truth change mask ($\mathbf{b} = 1$):

$$SC(\mathbf{y}, \hat{\mathbf{y}}|\mathbf{b}) = \frac{1}{|\mathcal{C}|} \sum_{c \in \mathcal{C}} \frac{|\{\mathbf{b} = 1\} \cap (\{\mathbf{y} = c\} \cap \{\hat{\mathbf{y}} = c\})|}{|\{\mathbf{b} = 1\} \cap (\{\mathbf{y} = c\} \cup \{\hat{\mathbf{y}} = c\})|}, \tag{5}$$

where $\mathcal{C}$ represents the set of semantic classes.

**Semantic Change Segmentation (SCS).** Finally, the SCS score is calculated as the arithmetic mean of the BC and SC scores, providing a unified metric for semantic change detection:

$$SCS = \frac{1}{2} \left( BC(\mathbf{b}, \hat{\mathbf{b}}) + SC(\mathbf{y}, \hat{\mathbf{y}}|\mathbf{b}) \right). \tag{6}$$

## A.4 VISUALIZATION RESULT

### A.4.1 RESULT IN LULC SCENARIO

Figure 7 presents the LULC semantic segmentation results of our proposed STSUN on the single-temporal LoveDA dataset (Figure 7 a) and the multi-temporal Dynamic EarthNet dataset (Figure 7 b). The method achieves strong performance across both, effectively navigating challenges such as the complex spectral-spatial features, multi-scale objects in LoveDA, and the inherent temporal variations within Dynamic EarthNet. This proficiency stems from STSUN's ability to perform unified representation and modeling of remote sensing data across spatial, temporal and spectral dimensions. Specifically, the STSUN enables harmonized encoding and feature fusion across these dimensions, while the LGWA mechanism efficiently captures both local details and global contextual information, crucial for accurate LULC delineation in diverse scenarios.

## A.5 COMPUTATIONAL COMPLEXITY ANALYSIS

To clarify whether the performance gains of STSUN stem from architectural efficiency or increased model capacity, we provide a comprehensive comparison of model parameters and computations on 512×512 image size. Table 12 details the number of parameters (Params) and floating-point operations (FLOPs) for STSUN and key baseline models.

As shown in the table, STSUN maintains a highly efficient profile with only 8.97 M parameters and 10.31 G FLOPs. In contrast, several baselines require significantly higher computational resources. While STSUN is slightly larger than the most lightweight baselines CaSaFormer, it remains in the same order of magnitude while outperforming them. This analysis confirms that the performance improvements of STSUN are primarily driven by our proposed unified modeling paradigm and efficient architectural design, rather than simply scaling up model capacity.

## A.6 POSITIONING RELATIVE TO REMOTE SENSING FOUNDATION MODELS

Recent advancements in remote sensing, such as RingMo-Agent (Hu et al., 2025) and Falcon (Yao et al., 2025), have demonstrated the efficacy of large-scale self-supervised pre-training for multimodal representation learning. We respectfully clarify that STSUN occupies a distinct methodological niche compared to these foundation models. While existing foundation models primarily focus on unifying input representations across disparate platforms or modalities, STSUN introduces a novel architectural strategy for the simultaneous unification of both input and output across Spatial-Temporal-Spectral dimensions. This full-pipeline unification allows for consistent modeling of complex dense prediction tasks that involve heterogeneous dimensions.

Furthermore, the proposed STS unification strategy is orthogonal to the scaling paradigm of foundation models. STSUN is not designed to compete with the massive parameter space of models

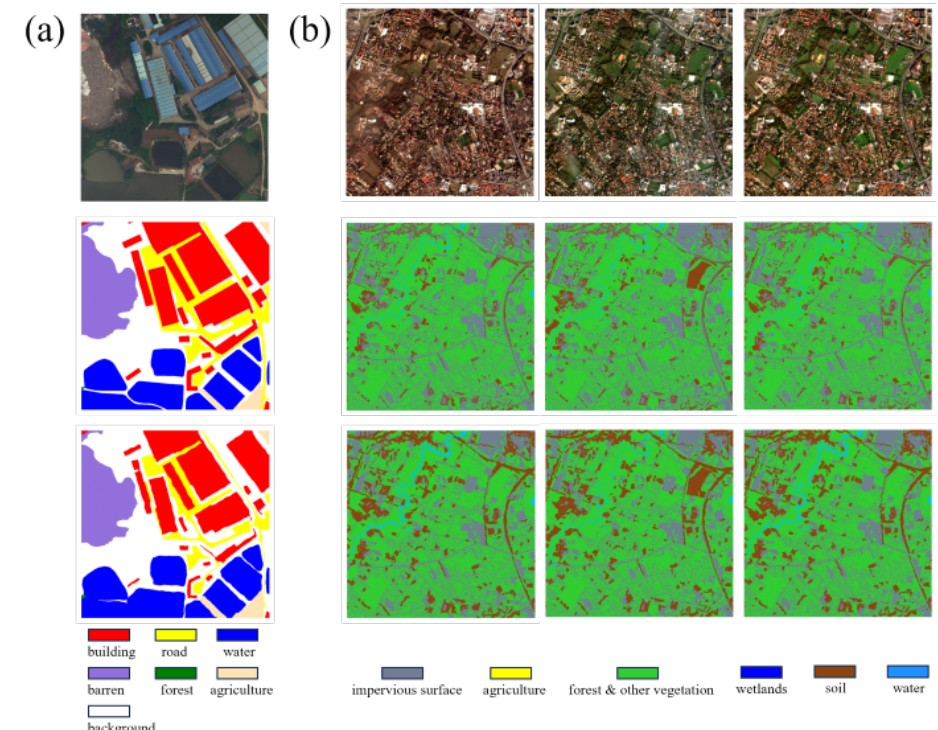

Figure 7: Sample inference results on for LULC scene datasets. The input images, ground truths and predictions are shown in the first, second and third rows, respectively. (a) LoveDA dataset sample. (b) DynamicEarthNet dataset sample.

Table 12: Comparison of model parameters and computational cost between STSUN and baseline models.

| Model | Params (M)↓ | FLOPs↓ (G) |
|---|---|---|
| CaSaFormer (Li et al., 2024a) | 4.44 | 1.99 |
| MRANet (Jiang et al., 2024) | 33.84 | 23.97 |
| AMTNet-50 (Liu et al., 2023) | 24.67 | 86.23 |
| TS-COUD (Zhao et al., 2024b) | 468.25 | 629.06 |
| ConvNext-V2 (Woo et al., 2023) | 89.00 | 223.90 |
| SSCBNet (Li et al., 2024b) | 28.73 | 72.49 |
| TSSCD (He et al., 2024) | 6.26 | 8.82 |
| STSUN | 8.97 | 10.31 |

like Falcon; rather, it offers a lightweight, fully supervised architectural solution. Consequently, our approach is highly compatible with the foundation model ecosystem. The STSUN architecture can effectively serve as a lightweight adapter or a specialized decoding head for pre-trained backbones, enabling general-purpose foundation models to adapt efficiently to specific multi-dimensional tasks without the prohibitive cost of full-model fine-tuning.

### A.7 EFFECT OF DATA UNIFICATION

To explicitly validate the effectiveness of our data unification strategy, we conducted a comparative ablation study using two Change Detection datasets: LEVIR-CD and TSCD. These datasets were specifically selected because they share identical task definitions and target categories but differ in

their data acquisition sources. This setup isolates the impact of multi-source data unification from task or category variations.

We established baselines by training our model independently on each dataset ($STSUN_{singe}$). We then compared these against a unified model ($STSUN_{unified}$) trained jointly on both datasets. As shown in Table 13, the unified model consistently achieves superior performance compared to the individually trained baselines on their respective test sets. This result demonstrates that our framework effectively mitigates the domain gap between heterogeneous data sources.

Table 13: Comparison over Data Unification on Two Building Datasets.

| Dataset | Methods | F1 (%)↑ | IoU (%)↑ |
|---------|---------|---------|----------|
| LEVIR-CD | $STSUN_{single}$ | 91.25 | 83.92 |
| | $STSUN_{unified}$ | **91.42** | **84.18** |
| TSCD | $STSUN_{single}$ | 65.98 | 49.23 |
| | $STSUN_{unified}$ | **66.25** | **49.52** |

### A.8 LIMITATIONS AND EXPECTATIONS

While the proposed Spatial-Temporal-Spectral Unified Network demonstrates considerable promise in harmonizing the analysis of heterogeneous remote sensing data, its current instantiation presents certain limitations which, in turn, illuminate clear and compelling trajectories for future research.

First, the operational flexibility of our framework is presently contingent upon the provision of explicit metadata at inference time, specifically in the form of predefined task and category embeddings. This requirement, while effective, curtails the model's autonomy and presupposes a level of a priori knowledge about the analytical objective. A significant advancement would be to imbue the model with the capacity to implicitly infer the task and desired output configuration directly from the contextual cues within the input data stream. Future work could explore methodologies grounded in meta-learning or employ sophisticated attention mechanisms that learn to dynamically weigh different aspects of the data, thereby deducing the analytical intent without explicit instruction.

Second, while our method achieves a critical unification at the data format and architectural input level across the spatial, temporal, and spectral dimensions, this does not inherently guarantee robust semantic generalization across the vast heterogeneity of remote sensing scenes, sensor modalities, and non-training semantic category sets. A promising path to surmount this limitation lies in elevating the STSUN framework into a large-scale, foundational model for Earth observation by employing advanced self-supervised or multi-modal pre-training strategies. The objective would be to produce a model that captures the fundamental structures of spatial-temporal-spectral data, enabling highly transferable representations that dramatically improve performance on a wide array of tasks with minimal, or even zero-shot, fine-tuning.

Third, the "flexible category set" capability is currently constrained by its reliance on a predefined, trainable library of class embeddings. Given the exclusive dependence on visual supervision within this closed set, the model lacks the inherent capacity for zero-shot generalization to truly unseen categories that are not included in the library. To bridge this gap, future iterations aim to integrate pre-trained text encoders to replace static embeddings with dynamic, language-aligned representations, thereby enabling authentic open-vocabulary recognition without the need for retraining.

Finally, the potential of our unified modeling paradigm has thus far been demonstrated exclusively within a supervised learning context. A significant opportunity exists to extend this framework into the self-supervised and vision-language domains. By pre-training the unified vision model on enormous, unlabeled remote sensing archives, one could construct a powerful Remote Sensing Vision Foundation Model. Such a model, pre-trained to comprehend the elemental structure of diverse STS data, could then be adapted with remarkable data efficiency for specialized downstream tasks. Concurrently, applying this unified input approach to Remote Sensing Vision-Language Models holds transformative potential. It would permit the training of these models on vast corpora of image-text pairs without the need for cumbersome, modality-specific engineering to handle varying spectral or temporal dimensions in the imagery. This would not only streamline the development and enhance the performance of RS-VLMs but also make them natively adaptable to the full diversity of remote sensing data, fostering a new generation of models capable of nuanced, cross-modal understanding of our planet.

