# OpenReview forum: "Spatial-Temporal-Spectral Unified Modeling for Remote Sensing Dense Prediction"
_ICLR.cc/2026/Conference — ICLR 2026 Conference Withdrawn Submission_

### Official Review · Reviewer_tKWs · 2025-10-18

**Soundness:** 4
**Presentation:** 3
**Contribution:** 4
**Rating:** 8
**Confidence:** 4

**Summary:**

This paper introduces a novel framework for remote sensing dense prediction. The core contribution is a unified model capable of handling heterogeneous inputs and outputs across spatial, temporal, and spectral dimensions. Furthermore, the model unifies three distinct dense prediction tasks and supports flexible semantic categories, therefore achieving data-task-class unification. The authors conduct experiments on seven diverse datasets of building and land use/land cover scenarios, demonstrating that their single unified model not only adapts to varied data-task-class configurations but also achieves state-of-the-art performance across all of them.

**Strengths:**

1.	The primary strength lies in its novel approach to unification. While prior work has used hypernetworks or other approaches for adaptive unification, it was typically limited to the channel dimension to handle multi-modal data [1-4]. This work innovatively connects the diversity across spatial, temporal, and spectral dimensions with the non-uniformity challenges in remote sensing concerning data, tasks, and classes. By unifying these core dimensions via a hypernetwork, the proposed method provides a natural and effective solution for a unified data-task-class framework.
2.	The decision to treat the temporal dimension separately, acknowledging its unique characteristics and designing a specific unification module for it, is an interesting and convincing design choice.
3.	The proposed method for unifying the spatial, temporal, and spectral dimensions of inputs and outputs appears to be general and highly adaptable, suggesting potential applicability to other research domains.
4.	The paper is supported by comprehensive and convincing experimental results. In particular, the ablation studies on multi-task and multi-class unification demonstrate the necessity and benefits of the proposed unified approach.
5.	The paper is well-written, clearly structured, and easy to follow.

[1] Xiong, Z., Wang, Y., Zhang, F., Stewart, A. J., Hanna, J., Borth, D., ... & Zhu, X. X. (2024). Neural plasticity-inspired multimodal foundation model for earth observation. arXiv preprint arXiv:2403.15356.
[2] Li, X., Li, C., Ghamisi, P., & Hong, D. (2025). Fleximo: A flexible remote sensing foundation model. arXiv preprint arXiv:2503.23844.
[3] Zhang, Y., Li, W., Zhang, M., Han, J., Tao, R., & Liang, S. (2025). SpectralX: Parameter-efficient domain generalization for spectral remote sensing foundation models. arXiv preprint arXiv:2508.01731.
[4] Sumbul, G., Xu, C., Dalsasso, E., & Tuia, D. (2025). SMARTIES: Spectrum-Aware Multi-Sensor Auto-Encoder for Remote Sensing Images. arXiv preprint arXiv:2506.19585.

**Weaknesses:**

1.	Some implementation details are not fully clear.
2.	The unification of each dimension is formalized with notation and some equations. However, the presentation could be made even clearer and more rigorous with a more comprehensive mathematical formulation.
3.	Moving some of the key visualization results from the appendix to the main paper would make the results more compelling and easier for the reader to interpret.
4.	It is recommended to tone down claims like "for the first time" in the introduction to avoid potential disputes and strengthen the paper's scholarly tone.

**Questions:**

1.	The proposed model not only achieves data-task-class unification but also outperforms the baseline models. What are the factors contributing to this superior performance?
2.	In Appendix A.1, the hypernetwork generates adaptive weights and biases for some dimensions, but only adaptive weights for others. What is the reasoning behind this design choice?
3.	The model requires explicit metadata as input. How could this metadata be inferred implicitly in future work to make the model more streamlined and user-friendly?
4.	This work primarily focuses on supervised models. What role could this unification framework play when extended to vision foundation models or large multi-modal models?

Minor Comments
1.	In Figure 2, consider adding "×N" to the encoder and decoder blocks to indicate that they are repeated.
2.	In Figure 3, the text should be ordered from top to bottom to maintain consistency with the flow in Figure 2.
3.	In Table 1, it would be helpful to include the size (e.g., number of samples) for each dataset.
4.	In Table 1, "Image Size" should be changed to "(H, W)" to be consistent with the dimensional notation used in the text.
5.	For all tables reporting results, consider adding arrows next to each metric to improve clarity, similar to the presentation in Table 8.
6.	The layout of some equations in Figure 5 needs adjustment for better readability.
7.	The positioning of Figure 6 and Figure 7 could be modified.

---

> ### Author Response · Authors · 2025-11-23
> **Response to tKWs (1/2)**
>
> We sincerely thank the reviewer for the encouraging summary and the positive assessment of our work ("excellent" scores in Soundness and Contribution). We are particularly grateful for the recognition of our novel unification approach across spatial, temporal, and spectral dimensions.
>
> To streamline our responses, we will refer to the Weaknesses as W1, W2, etc., the Questions as Q1, Q2, etc, and the Minor Comments as M1, M2, etc. Our replies will be marked as A1, A2, and so on. The revisions are all shown in red in the revised paper.
>
> > **[W1]**: Some implementation details are not fully clear.
>
> **[A1]** Thank you for pointing this out. In the revised Appendix **A.3 IMPLEMENTATION DETAILS**, we have added comprehensive explanations for the BC, SC, and SCS metrics. We clarified their definitions and calculation methods, as these metrics are specific to the DynamicEarthnet dataset evaluation protocol.
>
> > **[W2]**: The unification of each dimension is formalized with notation and some equations ...
>
> **[A2]** We appreciate this constructive suggestion. We have optimized the mathematical notation throughout the introduction and methodology section. Furthermore, we have introduced additional formulas to rigorously formalize the relationships between the spatial, temporal, and spectral dimensions, as well as the mapping between **dimensional unification** and the broader **data-task-class unification**. This makes the theoretical framework more robust and easier to follow.
>
> > **[W3]**: Moving some of the key visualization results from the appendix to the main paper would make the results ...
>
> **[A3]** We agree that visualizations are crucial for interpreting the results. Despite the space constraints, we have moved the qualitative visualization results on the **building datasets** from the Appendix to the **main text**. Figure 5 intuitively demonstrate the model's ability to handle dimensional unification and highlight its superior performance.
>
> > **[W4]**: It is recommended to tone down claims like "for the first time" in the introduction ...
>
> **[A4]** We accept this recommendation. We have carefully scanned the manuscript and removed claims such as "for the first time" to avoid potential disputes. We have rephrased these sentences to maintain a strictly scholarly tone while clearly highlighting our core contribution to dimensional unification in remote sensing.

---

> ### Author Response · Authors · 2025-11-23
> **Response to tKWs (2/2)**
>
> > **[Q1]**: The proposed model not only achieves data-task-class unification but also outperforms the baseline models ...
>
> **[A5]** The superior performance of STSUN primarily stems from three key factors:
>
> 1. **LGWA (Local-Global Window Attention):** Our proposed LGWA mechanism combines global attention with multi-oriented local window attention. This allows the model to effectively capture both long-range dependencies and fine-grained features in multiple directions, making it highly adaptable to remote sensing data with diverse spatial resolutions and coverages.
>
> 2. **Multi-Task Unification:** By unifying semantic segmentation, binary detection, and semantic change detection within a single architecture, the model benefits from **multi-task learning**. It leverages the intrinsic correlations between these tasks to jointly enhance feature representation and overall performance.
>
> 3. **Expanded Training Data:** Since STSUN achieves unification across data, tasks, and classes, we are no longer constrained to datasets with identical formats or label sets. This allows us to train on a much richer and more diverse collection of data, improving the model's generalization capability.
>
>
> > **[Q2]**: In Appendix A.1, the hypernetwork generates adaptive weights and biases for some dimensions ...
>
> **[A6]** This design choice is driven by the structural differences between input and output dimensional unification, specifically regarding the target dimensions. Taking the input and output time dimension unification shown in Figure 5(b) as an example:
>
> - **Input Temporal Unification (Weights + Bias):** When unifying the input time dimension, we map input features of shape $((L \times C), T_1)$ to a unified representation of shape $((L \times C), T)$. Here, the target dimension $T$ is a **static, predefined hyperparameter**. Therefore, the required bias (shape $T$) can be generated via a predefined static `[Cls]` token passed through a linear layer.
>
> - **Output Temporal Unification (Weights Only):** When unifying the output time dimension, we map the unified features $((L \times C), T)$ back to the output shape $((L \times C), T_2)$. Here, the target dimension $T_2$ is **dynamic**, depending on the specific input $T_1$ and the task requirements. Since $T_2$ varies, a static `[Cls]` token and linear layer cannot generate a bias of the correct dynamic shape. Consequently, we rely only on adaptive weights generated by the hypernetwork for this transformation.
>
>
> > **[Q3]**: The model requires explicit metadata as input. How could this metadata be inferred implicitly in future work ...
>
> **[A7]** In future work, we plan to incorporate a **text modality** to enable implicit metadata inference. While basic data attributes (spatial resolution, wavelengths, temporal length) may still require explicit metadata, **task types and semantic categories** can be inferred from natural language descriptions. Specifically, we propose using a Large Language Model (LLM) to encode a user's text description into a text embedding. This embedding would then be projected via separate linear layers to generate the **task embeddings** and **class embeddings**. This approach would significantly streamline the workflow, making the model more user-friendly by removing the need for manual task/class specification.
>
> > **[Q4]**: This work primarily focuses on supervised models ...
>
> **[A8]** The proposed dimensional unification strategy is highly transferable and can play a critical role in scaling up foundation models:
>
> 1. **Vision Foundation Models:** The strategy can unify the **Input Space** across spatial-temporal-spectral modalities. This enables a foundation model to accept inputs with arbitrary image sizes, spectral bands, and temporal lengths, projecting them into a unified visual representation for downstream processing.
>
> 2. **Multi-Modal Large Models:** The strategy can unify the **Visual Modality** for both understanding and generation. For instance, in a generative model, this framework would allow the model to perceive and generate remote sensing imagery with flexible configurations (e.g., generating a time-series of multi-spectral images) based on multimodal prompts.
>
>
> > **[M1-M7]**: In Figure 2, consider adding "×N" to the encoder and decoder blocks to indicate that they are repeated ...
>
> **[A9]** We thank the reviewer for their meticulous attention to detail. We have addressed all the minor comments (1-7) in the revised manuscript:
>
> 1. Added "×N" to encoder/decoder blocks in Figure 2.
>
> 2. Reordered text in Figure 3 to match the flow of Figure 2.
>
> 3. Included sample sizes for each dataset in Table 1.
>
> 4. Updated "Image Size" to "$(H, W)$" in Table 1 for consistency.
>
> 5. Added arrows to metrics in all results tables to indicate improvement direction.
>
> 6. Adjusted the layout of equations in Figure 5 for better readability.
>
> 7. Optimized the positioning of Figure 6 and Figure 7.

---

> > ### Comment · Reviewer_tKWs · 2025-11-24
> >
> > Thank the authors for their comprehensive and thoughtful rebuttal. The response has effectively addressed all my concerns.
> > Specifically, I appreciate the detailed clarifications regarding the hypernetwork design choices (Q2) and the insightful discussion on future implicit metadata inference (Q3). The revisions to the mathematical formulation and the decision to move key visualizations to the main text have improved the paper's clarity and presentation.
> > I believe the proposed STSUN framework makes a solid and novel contribution to the field by achieving effective data-task-class unification in remote sensing. Given the high quality of the work and the satisfactory resolution of my comments, I will raise my score to 10.

---

> > > ### Author Response · Authors · 2025-11-27
> > > **Thank You**
> > >
> > > We sincerely thank the reviewer for the positive feedback and the decision to raise the score to 10. We are greatly encouraged to learn that our clarifications regarding the hypernetwork design (Q2) and implicit metadata inference (Q3) have effectively addressed your concerns. We also appreciate your recognition of the STSUN framework as a solid and novel contribution to the field. We will ensure that the revised mathematical formulation and updated visualizations are preserved in the final version of the paper.

---

### Official Review · Reviewer_xSfp · 2025-11-01

**Soundness:** 3
**Presentation:** 2
**Contribution:** 3
**Rating:** 4
**Confidence:** 4

**Summary:**

The paper presents a unified framework addressing the challenge of inconsistent input and output configurations across different dense-prediction tasks in remote sensing. The proposed Spatial-Temporal-Spectral Unified Network (STSUN) integrates two key components: the Dimension Unified Module (DUM), which employs a transformer-based hypernetwork conditioned on metadata to adaptively map variable dimensions, and the Local-Global Window Attention (LGWA) module, which captures multi-scale contextual relationships. The model is designed to handle multiple dense-prediction tasks, including semantic segmentation, binary change detection, and semantic change detection, and can be trained in either single-task or multi-task settings, flexibly adapting to different spatial, temporal, and spectral domains.

**Strengths:**

The paper addresses an important and practical issue in remote sensing: heterogeneity of input and output structures. Encoding spatial, temporal, and spectral configurations as metadata is a smart, scalable idea. The local-global attention design helps handle multi-resolution dependencies, and multi-task training improves performance. Experiments are extensive, and results are strong across benchmarks.

**Weaknesses:**

1. Despite the strong empirical results, the core components are based on existing ideas. The DUM is a direct application of a transformer-based hypernetwork, and the LGWA is conceptually similar to other multi-scale windowed attention mechanisms found in models like the Swin Transformer or SegFormer.
2. The paper lacks theoretical analysis or deeper insight into its decoupled unification strategy, relying solely on experimental ablation (Table 11) to justify its design.
3. Although the appendix includes implementation details, the main exposition can be conceptually unclear. Notation is inconsistent, metadata definitions are vague, and the link between input and output dimensions must be inferred. These issues hinder a full understanding and make reimplementation challenging. For example, the notational system ($T_1$, $T_2$, $C_1$, $C_2$) is confusing, and metadata ($M_{in}$, $M_{out}$) lacks clear definition. Please clarify these structures and their correspondence.
4. The description of data dimensions (on page 1) and Figure 2 is overly verbose and could be condensed for readability.

**Questions:**

1. The "flexible category set" capability (Section 4.4) relies on selecting from a predefined and trainable set of class embeddings. Could the authors clarify the model's behavior with a truly 'new' or 'unseen' category not included in this predefined set? Would this scenario require retraining to add a new embedding, or can the model generalize in a zero-shot manner?
2. The main comparison tables (e.g., Tables 2-8) compare the STSUN_unified model (trained on combined data) against SOTA methods trained on single datasets. This makes it difficult to distinguish architectural benefits from the benefits of multi-task/multi-dataset training. Could the authors add the STSUN_single results (from Table 9) to these main tables for a more direct comparison against the SOTA baselines?
3. Please provide parameter counts or FLOPs for the STSUN model and the key baselines. This would help clarify whether the performance gains stem from the proposed architecture or from a significantly larger model capacity.

---

> ### Author Response · Authors · 2025-11-23
> **Response to xSfp (1/2)**
>
> We sincerely thank the reviewer for the constructive feedback and the positive assessment of our work's soundness, contribution, and the extensiveness of our experiments. We appreciate your recognition of our core idea regarding metadata encoding as "smart and scalable."
>
> To streamline our responses, we will refer to the Weaknesses as W1, W2, etc., and the Questions as Q1, Q2, etc. Our replies will be marked as A1, A2, and so on. The revisions are all shown in red in the revised paper.
>
> > **[W1]**: Despite the strong empirical results, the core components are based on existing ideas ...
>
> **[A1]** We partially agree with your observation that our modules build upon established concepts; however, we emphasize that our contribution lies in the specific adaptation and innovative design tailored for the unique challenges of unified remote sensing modeling.
>
> - **Regarding the Dimension Unified Module (DUM):** While inspired by hypernetworks, DUM innovatively integrates the hypernetwork paradigm with **dimension metadata**. We designed all the DUMs that unifiy the spatial-temporal-spectral dimension of input and output, and a specific unified strategy that decouples the temporal dimension. We unify dimensions in a strict sequence: Input Spatial-Spectral $\rightarrow$ Input/Output Temporal $\rightarrow$ Output Spatial-Spectral. This design is not a direct application but a specialized mechanism that enables the model to handle the extreme heterogeneity of remote sensing data, tasks, and categories within a single architecture.
>
> - **Regarding Local-Global Window Attention (LGWA):** While Swin Transformer and SegFormer utilize local windows to reduce computation, LGWA advances this by **integrating global attention with multi-shaped local attention**. Instead of relying solely on square windows, we designed diverse local window shapes to capture features in multiple directions simultaneously. This allows the model to efficiently extract both global context and multi-directional local nuances, which is critical for adapting to the diverse spatial resolutions and coverage areas inherent in remote sensing imagery.
>
>
> > **[W2]**: The paper lacks theoretical analysis or deeper insight into its decoupled unification strategy ...
>
> **[A2]** Thank you for this suggestion. We have strengthened the analysis in the Sec. 4.5 of revised manuscript to provide deeper insights into our design choices. Relevant literature has also been cited to support these design rationales.
>
> > **[W3]**: Although the appendix includes implementation details, the main exposition can be conceptually unclear ...
>
> **[A3]** Thank you for pointing this out. We have revised the notation system to ensure clarity and consistency throughout the paper.
>
> - We replaced the previous symbols and standardized the notation as follows: $T_1, C_1, (H_1, W_1)$ represent the input temporal, spectral, and spatial dimensions, respectively; $T_2, C_2, (H_2, W_2)$ represent the output dimensions. Correspondingly, metadata is denoted as $M_{T_1}, M_{C_1}, M_{(H_1,W_1)}$ and $M_{T_2}, M_{C_2}, M_{(H_2,W_2)}$.
>
> - We have also introduced explicit formulas to rigorously define the relationships between dimension unification and the unification of data, tasks, and categories. This revision makes the correspondence between input/output structures and metadata explicit and easy to follow.
>
>
> > **[W4]**: The description of data dimensions (on page 1) and Figure 2 is overly verbose ...
>
> **[A4]** We have refined the presentation as suggested.
>
> - **Page 1:** We condensed the textual description of data dimensions and utilized formal mathematical expressions to clearly define the connotation and connection of each dimension.
>
> - **Figure 2:** We streamlined the detailed caption and text describing the five stages of the model. The revised description now highlights the holistic architecture and the dimension unification strategy, rather than getting bogged down in repetitive stage-wise details.

---

> ### Author Response · Authors · 2025-11-23
> **Response to xSfp (2/2)**
>
> > **[Q1]**: The "flexible category set" capability (Section 4.4) relies on selecting from a predefined and trainable set of class embeddings ...
>
> **[A5]** Currently, since the model is trained via supervised learning using only visual modalities and limited-scale data, it **cannot generalize to unseen categories in a zero-shot manner**. The "flexible category set" capability refers to the model's ability to dynamically select any subset of categories from the *predefined* learnable embedding pool during inference, rather than generating embeddings for entirely new concepts.
> We have added a discussion on this in **Section A.8 (LIMITATIONS AND EXPECTATIONS)**. We clarify that this is a current limitation and propose future work to integrate text modalities to achieve true zero-shot generalization for new categories, such as replacing static embeddings with dynamic text-based embeddings via CLIP-like encoders.
>
> > **[Q2]**: The main comparison tables (e.g., Tables 2-8) compare the STSUN_unified model (trained on combined data) against SOTA methods trained on single datasets ...
>
> **[A6]** This is an excellent suggestion for a fairer comparison. We have added the experimental results of `STSUN_single` (trained on single datasets) to the comparison tables for the **4 building datasets**.
> The results demonstrate that **`STSUN_single` achieves SOTA-level performance**, yielding results comparable to the best specific baselines. This proves that our architecture is effective even without multi-task training. The simplified architecture of STSUN relies on the DUM and LGWA without other complex tricks, which explains why it performs similarly to SOTA in single-task settings. However, the significant gains observed in `STSUN_unified` confirm that the unified paradigm allows the model to leverage larger-scale data and multi-task joint learning to surpass single-task limitations.
>
> > **[Q3]**: Please provide parameter counts or FLOPs for the STSUN model and the key baselines ...
>
> **[A7]** We have added **Section A.5** to provide a detailed comparison of Parameters and FLOPs between STSUN and key baselines. Given the large number of baselines, we selected the best-performing method for each dataset as the key baseline for comparison.
> The comparison Table 12 shows that **STSUN is computationally efficient**, with parameters and FLOPs comparable to or lower than the baselines. This confirms that the performance improvements stem from the architectural design and unification strategy, rather than scaling up model capacity.

---

### Official Review · Reviewer_NX23 · 2025-11-02

**Soundness:** 2
**Presentation:** 2
**Contribution:** 2
**Rating:** 2
**Confidence:** 5

**Summary:**

This manuscript proposes the Spatial-Temporal-Spectral Unified Network (STSUN), a framework designed to achieve unified dense prediction across diverse remote sensing tasks and data configurations. The authors identify key limitations in existing deep learning models for remote sensing, including fixed input-output configurations, task-specific architectures, and rigid category sets, which hinder adaptability to heterogeneous data and multi-task scenarios.

**Strengths:**

1.	The paper is clearly written and easy to follow, with a well-structured presentation of the problem and proposed approach.
2.	The idea of unifying spatial, temporal, and spectral dimensions within a single framework is interesting and relevant to challenges in remote sensing dense prediction.

**Weaknesses:**

1.	The unification of spatial, temporal, and spectral dimensions has already been explored in several recent remote sensing foundation models, such as RingMo-Agent [1] and Falcon [2], which aim to build unified representations across multi-platform and multi-modal data. The paper does not discuss or compare its approach with these existing large-scale models, limiting the clarity of its novelty and positioning. [1] RingMo-Agent: A Unified Remote Sensing Foundation Model for Multi-Platform and Multi-Modal Reasoning [2] Falcon: A Remote Sensing Vision-Language Foundation Model
2.	The idea of flexible semantic class sets is not entirely novel. Prior works on open-vocabulary and open-set segmentation in remote sensing already address similar challenges. The paper does not discuss or position its proposed trainable category-embedding mechanism relative to these approaches, which reduces the clarity of its contribution in this context.
3.	The method relies on metadata and hypernetworks to generate adaptive linear layers for unifying arbitrary spatial, temporal, and spectral dimensions. In practice, the claimed “complete unification of STS dimensions” may be constrained by variations in spatial resolution, spectral coverage, and temporal sampling intervals. The authors do not provide sufficient experimental validation to support the generalizability of this approach.
4.	The feasibility of the Temporal Unified Module (TUM) is unclear. TUM fuses multi-temporal features using hypernetworks and metadata, mapping them to arbitrary output temporal lengths. For high-temporal-resolution change detection or long sequence data, such linear mappings may not adequately capture complex temporal dynamics, potentially causing information loss or performance degradation. The paper does not include experiments on long temporal sequences or ablation studies to validate TUM’s effectiveness.
5.	The Local-Global Window Attention (LGWA) module uses multiple local windows of predefined shapes alongside a single global window to capture features at different scales. Fixed window sizes and shapes may not adapt well to varying spatial resolutions or object scales. When input data vary substantially, for example in satellite type, spatial resolution, or spectral channels, this strategy may lead to unstable performance. Furthermore, no experiments are provided comparing LGWA with other adaptive attention mechanisms such as Swin Transformer or CSWin.
6.	The experimental results show that the proposed method’s performance is not particularly strong. Compared with existing state-of-the-art methods for building extraction and building change detection, the accuracy exhibits a noticeable gap. Additionally, the paper does not include comparisons with large foundation models such as the Segment Anything Model or remote sensing models based on SAM, which would help contextualize the method’s practical effectiveness.
7.	The ablation studies and analysis are limited, making it difficult to fully support the authors' claim of achieving unification across arbitrary spatial, temporal, and spectral dimensions. More comprehensive experiments are needed to demonstrate the contribution of each component and to validate the generalization of the proposed framework.

**Questions:**

1.	Relation to existing foundation models: Could the authors clarify how STSUN differs from recent remote sensing foundation models such as RingMo-Agent and Falcon? Have the authors considered including a comparison or discussion of these models to better position the novelty of their approach?
2.	Flexible semantic class sets: How does the proposed trainable category-embedding mechanism compare with prior open-vocabulary or open-set segmentation approaches in remote sensing? Could the authors provide experiments or analysis to demonstrate the advantage of their method over these existing paradigms?
3.	STS dimension unification: The method relies on metadata and hypernetworks to unify spatial, temporal, and spectral dimensions. Can the authors provide more empirical evidence to show that this approach generalizes across varying spatial resolutions, spectral coverage, and temporal sampling intervals? For example, have they tested the model on datasets with highly heterogeneous input configurations?
4.	Temporal Unified Module (TUM): For long temporal sequences or high-temporal-resolution change detection, how does TUM handle complex temporal dynamics? Could the authors include ablation studies or experiments on longer sequences to validate the effectiveness and stability of TUM?
5.	Local-Global Window Attention (LGWA): How sensitive is LGWA to the choice of local window sizes and shapes, particularly when input data vary in spatial resolution, object scale, or spectral channels? Have the authors compared LGWA with other adaptive attention mechanisms such as Swin Transformer or CSWin to verify its effectiveness?
6.	Experimental performance and comparisons: The current experiments show a noticeable gap in accuracy compared with state-of-the-art building extraction and change detection methods. Could the authors provide comparisons with foundation models such as the Segment Anything Model or RS models based on SAM to better contextualize the performance?
7.	Ablation studies: The current ablation experiments appear limited. Could the authors provide more detailed component-level analyses to demonstrate the contribution of each module and to support their claim of achieving full unification across arbitrary spatial, temporal, and spectral dimensions?

---

> ### Author Response · Authors · 2025-11-23
> **Response to NX23 (1/2)**
>
> We sincerely thank the reviewer for the insightful comments and the time dedicated to reviewing our manuscript. We are encouraged by the positive feedback regarding the clear presentation and the relevance of unifying spatial, temporal, and spectral dimensions. Below, we address the concerns raised in the Weaknesses and Questions sections point-by-point.
>
> To streamline our responses, we will refer to the Weaknesses as W1, W2, etc., and the Questions as Q1, Q2, etc. Our replies will be marked as A1, A2, and so on. The revisions are all shown in red in the revised paper.
>
> > **[W1, Q1]**: The unification of spatial, temporal, and spectral dimensions has already been explored ...
>
> **[A1]** We appreciate the opportunity to clarify the positioning of our work. We respectfully wish to highlight distinct differences between STSUN and the foundation models mentioned:
>
> 1. **Scope of Unification:** While **RingMo-Agent** builds unified representations for multi-platform/multi-modal data, **Falcon** is primarily trained and tested on high-resolution RGB data. To the best of our knowledge, existing foundation models typically unify *either* the input spatial dimension *or* the input spectral dimension. In contrast, the core contribution of STSUN is the simultaneous unification of **both input and output** across the Spatial-Temporal-Spectral dimensions.
>
> 2. **Training Paradigm:** It is crucial to note that comparing STSUN directly with these large-scale models is methodologically difficult. STSUN is a fully supervised small model trained on specific visual datasets, whereas large foundation models utilize massive self-supervised pre-training, and agents integrate many large and small models as tools. STSUN is not designed to compete with the scale of foundation models but to propose a novel **architectural strategy** for unification.
>
>
> **Action:** To clarify this, we have added **Appendix A.6**. This section discusses how the proposed STS unification strategy is orthogonal to and compatible with visual foundation models, potentially serving as a lightweight adapter to enhance large models in multi-task scenarios.
>
> > **[W2, Q2]**: The idea of flexible semantic class sets is not entirely novel ...
>
> **[A2]** We would like to clarify the fundamental difference between our approach and Open-Vocabulary Segmentation (OVS):
>
> 1. **Modality Difference:** OVS methods typically rely on **text encoders** (e.g., CLIP) to encode category names, requiring substantial image-text paired data.
>
> 2. **Our Approach:** STSUN achieves flexible semantic prediction using a **purely visual modality**. We utilize predefined category embeddings and hypernetworks to adapt the classification head. This eliminates the dependency on textual descriptions or language models.
>
>
> Therefore, we believe a direct comparison with OVS methods is not entirely applicable, as the underlying assumptions and data requirements differ significantly.
>
> **Action:** We have revised **Section 2.3 (Flexible Semantic Class Sets)** to explicitly articulate these differences in data modality and implementation mechanisms.
>
> > **[W3, Q3]**: The method relies on metadata and hypernetworks to generate adaptive linear layers for unifying arbitrary spatial ...
>
> **[A3]** We agree that validation across diverse configurations is essential. To robustly verify the effectiveness of our metadata-driven unification, we carefully selected experimental datasets with high heterogeneity. Our experiments cover:
>
> - **Spatial Resolution:** Ranging from **0.075m to 10m**.
>
> - **Spectral Coverage:** Ranging from **3 bands (RGB)** to **13 bands (Sentinel-2)**.
>
> - **Temporal Sampling Interval:** Ranging from **1 day to 14 years**.
>
>
> The consistent performance of STSUN across these vastly different configurations provides strong empirical evidence that our metadata/hypernetwork approach effectively generalizes to variations in spatial, spectral, and temporal domains.
>
> **Action:** We have added a "Interval" column to **Table 1** and expanded the description in **Section 4** to highlight the diversity of the datasets used.
>
> > **[W4, Q4]**: The feasibility of the Temporal Unified Module (TUM) is unclear. ...
>
> **[A4]** The Temporal Unified Module (TUM) is designed to align input and output temporal dimensions, enable feature-level fusion for long-term image sequences [1], and handle complex dynamics by conditioning on temporal metadata.
>
> Regarding the concern about long sequences, our experimental setup specifically includes datasets where the temporal length varies from **1 to 24 time steps**, and the time span ranges from single points to **14 years**. This covers both high-temporal-resolution scenarios and long-term changes.
>
> The results demonstrate that TUM successfully handles these varying dynamics, validating its capability to capture complex temporal information via hypernetwork-generated weights.

---

> ### Author Response · Authors · 2025-11-23
> **Response to NX23 (2/2)**
>
> > **[W5, Q5]**: The Local-Global Window Attention (LGWA) module uses multiple local windows of predefined shapes ...
>
> **[A5]** LGWA is designed with multiple local windows of *different shapes* specifically to extract features across various directions and scales, mitigating the limitation of a single fixed size.
>
> **Action:** To empirically prove the superiority of LGWA:
>
> - We have included a comparison with **Swin Transformer’s Window Attention** in our ablation studies.
>
> - The results demonstrate that LGWA outperforms standard window attention, confirming its adaptability to diverse spatial resolutions and object scales.
>
>
> > **[W6, Q6]**: The experimental results show that the proposed method’s performance is not particularly strong ...
>
> **[A6]** We wish to clarify two points regarding performance and comparisons:
>
> 1. **Comparison with SOTA:** We have compared STSUN against a comprehensive suite of methods, including CNN-based, Transformer-based, and Hybrid architectures. This includes both classic stable baselines (e.g., BIT, SNUNet) and recent SOTA models (e.g., CaSaFormer, TS-COUD). We believe this benchmarking is extensive and demonstrates STSUN's competitive performance.
>
> 2. **Comparison with SAM:** As mentioned in [A1], comparing a supervised model trained on small-scale data (STSUN) with a zero-shot foundation model trained on 1 billion masks (SAM) is inherently unfair. Furthermore, standard SAM does not natively handle multi-temporal/spectral remote sensing data without substantial adaptation.
>
>
> **Action:** We added the **Appendix A.6** to clarify that, we position STSUN as a specialized unification strategy that can potentially be integrated with foundation models, rather than a direct competitor to zero-shot segmentation models.
>
> > **[W7, Q7]**: The ablation studies and analysis are limited, making it difficult to fully support the authors' claim ...
>
> **[A7]** We have conducted extensive experiments to validate the contribution of each component towards the goal of arbitrary STS unification:
>
> 1. **Data Diversity:** As detailed in Response 3, the extreme variance in our dataset parameters (Resolution: 0.075m-10m; Image Size: 120-1024; Bands: 3-13; Time Steps: 1-24) serves as intrinsic validation of the unification strategy.
>
> 2. **Component Ablation:** Section 4.5 validates the dimension unification strategy and LGWA (including the new Swin comparison). Sections 4.3 and 4.4 validate Task and Category unification.
>
> 3. **Data Unification Validation:** To explicitly prove the benefit of *unifying* datasets, we added **Appendix A.7**.
>
>   - **Experiment:** We utilized LEVIR-CD and TSCD datasets, which differ only in data source, sharing tasks/classes.
>
>   - **Result:** We compared a unified model (`STSUN_unified`) against models trained separately (`STSUN_levircd` and `STSUN_tscd`). The unified model achieved superior performance, conclusively demonstrating the benefits of our data unification approach.
>
>
> [1] Chen, P., Zhang, B., Hong, D., Chen, Z., Yang, X., & Li, B. (2022). FCCDN: Feature constraint network for VHR image change detection. *ISPRS Journal of Photogrammetry and Remote Sensing*, *187*, 101-119.

---

> > ### Comment · Reviewer_NX23 · 2025-11-25
> >
> > 1. The core contribution of this work lies in the simultaneous unification of spatial-temporal-spectral dimensions for both the input and output. This stands in contrast to foundation models and other existing algorithms, which typically unify dimensions solely at the input level. However, the fundamental advantage of this full unification remains unclear. While the authors demonstrate performance improvements empirically, they fail to adequately justify the underlying significance or rationale for unifying the output dimensions. I suggest the authors provide a more comprehensive discussion or supplementary experiments to substantiate the necessity and value of this core design choice.
> > 2. The distinction between the proposed 'flexible semantic class' and existing Open-Vocabulary (OV) semantic segmentation remains insufficiently elucidated. While standard OV approaches typically derive class embeddings via CLIP, the proposed method utilizes predefined embeddings generated through a purely visual modality, thereby eliminating dependency on a text encoder. However, the specific advantage of this design choice is not adequately justified. Although the authors claim it allows for 'efficient adaptation within the visual domain,' no experimental results or theoretical proofs are provided to substantiate this assertion.
> > 3. While the performance improvement yielded by the window self-attention mechanism is evident, the authors have not conducted experiments to determine whether the model's performance is sensitive to window size settings. Furthermore, it remains unclear whether the model maintains stability when a fixed window size is applied to data exhibiting significant variations in scale and spatial resolution. Sensitivity analyses addressing these aspects are missing.
> > 4. The setup of the ablation studies remains incomplete. Specifically, the individual effectiveness of the proposed components—namely ISSUM, TUM, and OSSUM—has not been verified through component-wise ablation experiments. A detailed breakdown of each module's contribution to the final performance is required.

---

> > > ### Author Response · Authors · 2025-12-01
> > > **Response to NX23**
> > >
> > > We thank the reviewer for the constructive response. Below, we address the concerns point-by-point.
> > >
> > > **A1:** We apologize for not making this rationale sufficiently clear in the submission. As emphasized in the Introduction and Method sections, the unification of output dimensions is intrinsically mapped to the unification of tasks, and categories:
> > >
> > > 1. **Temporal Output:** Correlated with the **type of dense prediction tasks** (Task Unification).
> > >
> > > 2. **Spectral Output:** Correlated with the **set of semantic categories** (Category Unification).
> > >
> > >
> > > Therefore, the core significance of unifying output dimensions lies in enabling a single model to perform diverse dense prediction tasks and predict across multiple semantic category sets simultaneously. To substantiate this validity:
> > >
> > > - **Task Unification (Sec. 4.3):** Results demonstrate that our unified output design not only supports multi-tasking but also leverages multi-task learning to jointly improve performance across individual tasks.
> > >
> > > - **Category Unification (Sec. 4.4):** Results show that a single model predicting multiple category sets achieves performance comparable to task-specific models while significantly improving category prediction efficiency.
> > >
> > >
> > > **A2:** We appreciate this opportunity to clarify. We address this from three perspectives:
> > >
> > > - **a) Clarification of Research Scope:** Our primary objective is **Unified Modeling** (unifying heterogeneous spatial-temporal-spectral dimensions for data, tasks, and categories), rather than Open-Vocabulary Semantic Segmentation. Consequently, a direct comparison with OV methods is neither the primary focus nor strictly necessary for this scope. We acknowledge the limitation regarding unseen classes in **Section A.8** and outline plans to incorporate text encoders for OV capabilities in future work.
> > >
> > > - **b) Empirical Evidence:** The efficiency of our visual-based semantic category embeddings is extensively validated. As shown in **Sec. 4.1, 4.2, and 4.4**, extensive experiments across **7 diverse datasets** demonstrate that, guided by these embeddings, our model flexibly predicts multiple category sets and achieves state-of-the-art (SOTA) performance, outperforming all comparison methods.
> > >
> > > - **c) Theoretical Rationale:** Theoretically, trainable dynamic semantic embeddings are often superior to pre-trained static text embeddings (e.g., CLIP) for adapting to specific remote sensing datasets. For instance, textual descriptions often harbor ambiguity regarding scale. The class "Forest" in high-resolution data corresponds to single tree features, whereas in medium-resolution data, it corresponds to texture patches. Static text embeddings struggle to bridge this domain gap [1]. In contrast, our proposed embeddings assign distinct, learnable vectors to these datasets, dynamically adapting to the feature discrepancies of the same semantic concept across different resolutions.
> > >
> > >
> > > **A3:** Thank you for the suggestion. We have conducted additional analyses:
> > >
> > > - **a) Sensitivity Analysis:** To investigate the impact of window size, we performed an ablation study comparing our "Mixed" window strategy against fixed window shapes (Strip, Square, Column). The results, presented in the table below, indicate that the Mixed strategy yields the best performance, verifying the effectiveness of utilizing multi-shaped local windows.
> > >
> > > | **Window Size** | **F1(%) ↑** | **IoU(%) ↑** |
> > > | --- | --- | --- |
> > > | Strip | 65.74 | 48.97 |
> > > | Square | 65.97 | 49.21 |
> > > | Column | 65.78 | 49.02 |
> > > | **Mixed (Ours)** | **66.48** | **49.79** |
> > >
> > > - **b) Stability Verification:** Our experiments already cover datasets with significant variations in scale (image sizes from $120\times120$ to $1024\times1024$) and spatial resolution ($0.075m$ to $10m$). The consistent SOTA performance across these heterogeneous settings (as reported in the paper) serves as strong evidence of the model's stability and robustness to scale variations.
> > >
> > > **A4:** These components serve strictly functional roles designed to enable the unification of Data, Tasks, and Categories through the Spatial-Temporal-Spectral dimensions. Therefore, we verified their contributions by ablating the specific unification capabilities they enable:
> > >
> > > - **Data Unification:** Analyzed in **Sec. A.7**. Results show it allows adaptation to diverse inputs while boosting performance.
> > >
> > > - **Task Unification:** Analyzed in **Sec. 4.3**. Results prove it allows simultaneous multi-task application with mutual performance benefits.
> > >
> > > - **Category Unification:** Analyzed in **Sec. 4.4**. Results demonstrate it maintains high accuracy while significantly enhancing prediction efficiency.
> > >
> > >
> > > These experiments collectively validate the necessity and effectiveness of each proposed module.
> > >
> > > [1] An, K., Wang, Y., & Chen, L. (2025). Soft-Guided Open-Vocabulary Semantic Segmentation of Remote Sensing Images. IEEE Transactions on Geoscience and Remote Sensing.

---

### Official Review · Reviewer_Wnht · 2025-11-25

**Soundness:** 1
**Presentation:** 1
**Contribution:** 2
**Rating:** 2
**Confidence:** 5

**Summary:**

The authors propose the Spatial-Temporal-Spectral Unified Network (STSUN) to leverage diverse dimensions of input data for dense downstream tasks. The proposed method uses metadata to learn representations from diverse spatial resolutions, temporal lengths, and spectral bands.

The overall framework aims to integrate diverse dense tasks within a single architecture using task embeddings that support arbitrary class subsets. It is composed of fives proposed modules: the Input Spatial-Spectral Unified Module (ISSUM) extracts spatial-spectral embeddings, the Local-Global Window Attention mechanism (LGWA) learns multi-spatial patterns, the Temporal Unified Module (TUM) homogenizes the temporal dimension, the Decoder Local-Global Attention Blocks processes the global embedding and the Output Spatial-Spectral Unified Module (OSSUM) generates the outputs.

The authors conducted exhaustive experiments on building and land use land cover (LULC) scenarios, comparing an extensive list of competing methods, including a version of their model trained on a single dataset and ablation studies to partially justify their approach.

**Strengths:**

I acknowledge that this review has been produced considering the revised manuscript.

1. The framework, which incorporates metadata of data inputs and dense prediction tasks, is well introduced and carefully distinguishes the spatial, temporal, and spectral input and output dimensions.

2. The authors leveraged metadata to generate an adapted architecture for input and output dimensions, proposing five modules to unify their representations.

3. The learnable task embedding effectively guides the model to perform a given task within the set of possible dense tasks.

4. The category embedding set enables flexible predictions across predefined semantic categories.

5. The experiments are comprehensive, including a relevant list of competing methods specific to each dataset and scenario, an ablation study of several proposed modules, and an analysis of single-dataset versus multi-dataset training of their method.

**Weaknesses:**

I acknowledge that this review has been produced considering the revised manuscript.

1. The limitations that this work addresses (L.90-103) are partially studied and grounded; however, several nuances should be considered, which weaken the motivations:
 a. Fixed configurations: ViTs are capable of processing sequences of arbitrary length; if spatial-temporal-spectral cubes are divided into tokens, this is no longer a limitation in theory [1, 2, 3, 4, 5].
 b. Fixed task: I would like to quote a recent work by Siméoni et al. [6], which aligns with recent remote sensing literature: "In particular, SSL produces rich, high-quality visual features that are not biased toward any specific supervision or task, thereby providing a versatile foundation for a wide range of downstream applications." Since most remote sensing foundation models and generalist backbones achieving state-of-the-art performance align with such statements, one may question whether learning correlations between the mentioned tasks is necessary, or experimental comparisons against SSL-pretrained models should be provided to support this argument.
 c. Fixed set of categories: Remote sensing foundation models and generalist backbones are designed to be adapted to dense tasks with a single linear layer, which may be simpler than the authors' proposed method (see W.4.c). For example, species distribution models, well established in remote sensing, produce a distribution of classes per pixel, ranging up to 10,000 classes, making these approaches nearly agnostic to additional classes [7].

2. There is insufficient related work on spatial-temporal-spectral methods for remote sensing pretraining and applications, such as S2MAE [2], SkySense++ [4], Galileo [5], all demonstrating strong performance on dense tasks with task-specific heads without requiring cross-task correlation training. Additionally, the differences between the proposed method and TSViT [7] are unclear, as their method efficiently processes spectral, spatial, and temporal information in a similar sequence, as illustrated in Figure 2 and explained in Section 3.2. A comprehensive comparison of the proposed method with existing remote sensing domain methods would be appreciated.

3. Despite the generalist formulation and architectural adaptability through hypernetworks to create adapted linear layers, once the model is instantiated with fixed dimensions (Eq. 3), it cannot accommodate different dimensions, such as shorter time series or different numbers of spectral bands, due to the fixed size of linear layers used for projections (Figure 6). Although the core transformations operate within unified dimensions, linear layers must still be adapted for each module to change dimensions across datasets, which could be reframed as modality-specific or task-specific tokenizers, as used in remote sensing foundation models.

4. It is unclear why foundation models, essential methods in remote sensing nowadays, were excluded from this study despite the authors' statements in A.6. Remote sensing foundation models are not restricted to unifying input representations, as mentioned in A.6; they aim to learn generalist representations that perform better on any downstream task, including all dense prediction tasks and many others (see the quote from Siméoni et al. in W.1.b).
a. The authors state that their method aims to better unify input and output dimensions, which could be interpreted as learning generalist representations that map any input and output dimensions. This is precisely the purpose of remote sensing foundation models, and thus both approaches address the same research direction.
b. The authors also state that "STSUN is not designed to compete with the massive parameter space of models like Falcon"; however, they compared their method to models with parameter counts ranging from 6.26M to 468.25M (Table 12). Most remote sensing foundation models operate in the same range by proposing multiple backbone versions, e.g., Galileo (0.8M to 85M) [5], OlmoEarth (1.4M to 300M) [9], AnySat (125M to 128M) [10], S2MAE (86M to 632M) [2], and others [1, 3, 11] that would be suitable for fair comparison.
c. Foundation models are designed to be either fully fine-tuned or fine-tuned with a single linear layer to perform any dense task. Since the proposed OSSUM requires two linear layers and a transformer block to project the class dimension, plus an additional layer to project to the output space, comparison with foundation models should be considered fair.

5. There is insufficient comparison with Perceiver IO [12], which demonstrates a very similar method based on attention fusion to unify input dimensions through linear transformations and learns the global output tensor carrying task information, which is generated with an output query.

6. The proposed Local-Global Window mechanism is not novel, as it was introduced in Swin Transformers [13].

7. There is no LULC scenario with time series, despite this being one of the most common use cases, particularly for agricultural applications. The PASTIS [14] dataset would be a well-suited benchmark for this study.

8. STSUN appears to be the only model trained on six datasets, whereas competing methods appear to be trained only on the dataset of interest, making direct comparison unfair. Training a similar method such as TSViT [8] or Perceiver IO [12] on multiple datasets would better assess whether performance gains result from the proposed architecture or from the combination of datasets used in training. Note that fine-tuning a foundation model on each dataset would be more appropriate, as they have been pretrained on larger datasets than competing methods.

9. The experiments lack standard error estimates evaluating the variability of the proposed method (e.g., across different initialization seeds), which raises questions about the significance of numerically similar results. For example, it is difficult to conclude that the "category unification" strategy is relevant based on Table 10 results, given the very close numerical values that could fall within the same standard error range.

10. There is a lack of ablation study to justify the selection of all proposed modules.

11. The overall paper is difficult to read due to the introduction of numerous proposed concepts, some of which are not novel, and a notation framework that is initially well introduced but subsequently difficult to follow because of exhaustive naming of modules and layers that is not necessary. Figure 6 should be improved and referenced correctly to help readers understand each module and how they interact. Note that several layer and module formulations are redundant; the submission would benefit from consolidating them into a more generalist concept.

## References:

[1] Xiong et al., Neural Plasticity-Inspired Multimodal Foundation Model for Earth Observation. In ArXiv 2024.

[2] Li et al., S2MAE: A Spatial-Spectral Pretraining Foundation Model for Spectral Remote Sensing Data. In CVPR 2024.

[3] N. Bountos et al., FoMo: Multi-Modal, Multi-Scale and Multi-Task Remote Sensing Foundation Models for Forest Monitoring. In AAAI 2025.

[4] K. Wu et al., A semantic-enhanced multi-modal remote sensing foundation model for Earth observation. In Nature machine intelligence 2025.

[5] G. Tseng et al., Galileo: Learning Global & Local Features of Many Remote Sensing Modalities. In ICML 2025.

[6] Siméoni et al., DinoV3. In ArXiv 2025.

[7] Zbinden et al., MaskSDM: Adaptive Species Distribution Modeling Through Data Masking. In ECCV Workshop.

[8] Tarasiou et al., ViTs for SITS: Vision Transformers for Satellite Image Time Series. In CVPR 2023.

[9] Herzog et al., OlmoEarth: Stable Latent Image Modeling for Multimodal Earth Observation. In ArXiv 2025.

[10] G. Astruc et al., AnySat: One Earth Observation Model for Many Resolutions, Scales, and Modalities. In CVPR 2025.

[11] A. Fuller et al., CROMA: Remote Sensing Representations with Contrastive Radar-Optical Masked Autoencoders. In NeurIPS 2023.

[12] Jeagle et al., Perceiver IO: A General Architecture for Structured Inputs & Outputs. In ICLR 2022.

[13] Liu et al., Swin Transformer: Hierarchical Vision Transformer using Shifted Windows. ICCV 2021.

[14] Sainte Fare Garnot et al., Panoptic Segmentation of Satellite Image Time Series with Convolutional Temporal Attention Networks. In ICCV 2021.

**Questions:**

## Questions
1. Considering the details provided in W.1, to what extent does unifying dense tasks provide benefits compared to using task-specific, parameter-efficient heads? One would expect experimental evidence demonstrating that learning task correlations improves results compared to existing methods.

2. Considering the details provided in W.1 and W.4, to what extent does a fixed set of classes introduce rigidity in the generalization of large backbones, given that a simple linear layer can be fine-tuned for the final task? One would expect experimental evidence demonstrating that using category embedding sets improves results compared to a plug-and-play approach.

3. What are the key differences between the proposed approach and a foundation model with one linear layer per dataset to perform each dense task simultaneously?

4. Since the proposed method depends on input and output dimensions (Section 3.1), do all linear layers change with respect to each dataset used in training? If so, is it correct that one must train dataset-specific linear layers?

5. What are the differences between the proposed method and Perceiver IO [12]? Why was Perceiver IO not considered as a competing method, given that its overall goal is to unify any input dimensions and output tasks?

6. How are the category embedding set and selected category subsets defined? Is the method still limited to a fixed number of categories, as described in the limitations?

References are listed in the Weaknesses section.

## Comments

- Section 3.2 lacks references to the architecture details of each module (Appendix A.1). This section is important to understand the methodology while being difficult to read and follow all the steps.
- Figure 6: this crucial figure is confusing; it lacks links between the modules to better understand how dimensions match between each module.
- Figure 6: to what corresponds to the list of numbers related to the spectral wavelength input in (a)?
- Table 1: please explain all acronyms in the caption.

---

> ### Author Response · Authors · 2025-11-30
> **Response to Wnht (1/5)**
>
> We sincerely thank the reviewer for the constructive feedback. To streamline our responses, we will refer to the Weaknesses as W1, W2, etc., the Questions as Q1, Q2, etc., the Comments as C1, C2, etc. Our replies will be marked as A1, A2, and so on. The revisions are all shown in red in the revised paper.
>
> > **[W1]**: The limitations that this work addresses (L.90-103) are partially studied and grounded ...
>
> **[A1]** We appreciate your detailed insights. We clarify our design choices below:
>
> a. Regarding Fixed Configurations:
>
> While we acknowledge that ViTs handle sequences of arbitrary lengths via tokenization, simply dividing spatial-temporal-spectral cubes into tokens is often sub-optimal or inapplicable for Remote Sensing (RS) data due to physical and scale discrepancies:
>
> 1. **Spectral Incompatibility:** RS spectral bands carry distinct physical meanings (e.g., RGB vs. Near/Mid/Far-Infrared). A fixed tokenization strategy often ignores these semantic differences. Our method explicitly encodes spectral wavelengths as metadata, allowing the model to distinguish and process bands based on their physical properties rather than just channel count.
>
> 2. **Spatio-Temporal Heterogeneity:** RS data exhibits massive variations in spatial resolution (e.g., 0.075m to 10m in our datasets) and temporal sampling (e.g., daily vs. decadal intervals in our datasets). Direct tokenization of fixed spatial patches or time windows fails to capture this granularity. In contrast, our metadata-driven encoding allows the network to adaptively process these variances.
>
> 3. **Backbone Flexibility:** Unlike tokenization strategies that rely heavily on Transformer backbones, our unified strategy is backbone-agnostic, offering greater flexibility.
>
>
> b. Regarding Fixed Tasks:
>
> We agree Foundation Models (FMs) produce versatile features. However, the robust features learned by FMs and the multi-task learning (MTL) approach of STSUN are complementary, not mutually exclusive.
>
> - MTL can enhance FMs: Integrating unified task modeling (as proposed in STSUN) into FMs allows a single model to handle multiple downstream tasks simultaneously without managing separate task-specific heads.
>
> - Synergy: Recent literature demonstrates that MTL objectives during pre-training or fine-tuning can further improve the robustness and generalizability of foundation models [1, 2].
>
>
> c. Regarding Fixed Categories:
>
> Similarly, FM robustness and our flexible category prediction are complementary.
>
> - Current FMs typically require fine-tuning separate linear heads for different sets of semantic categories.
>
> - In contrast, our **Dimension Unification Module** utilizes trainable category embeddings as guidance. This allows a single STSUN model (after a single fine-tuning session) to predict across multiple disparate category sets dynamically, which is more efficient than maintaining distinct heads for every new dataset or classification scheme [3].
>
>
> > **[W2]**: There is insufficient related work on spatial-temporal-spectral methods for remote sensing pretraining and applications ...
>
> **[A2]** Thank you for highlighting these references.
>
> 1. **Related Work:** We will update Section 2 to include a more comprehensive discussion on recent RS Foundation Models. We will explicitly analyze how they process input data to better highlight the unique advantages and contributions of our Spatial-Temporal-Spectral unified strategy.
>
> 2. **Comparison with TSViT:** While TSViT and STSUN both process dimensions sequentially, their objectives and mechanisms differ significantly:
>
>   - **TSViT** targets crop classification using long time-series data with low spatial resolution. It prioritizes temporal fusion first to aggregate classification features, followed by spatial refinement.
>
>   - **STSUN** targets unified dense prediction across heterogeneous inputs. We isolate the temporal dimension to ensure adaptability. Our processing sequence is: (1) Input Spatial-Spectral unification → (2) Input/Output Temporal unification → (3) Output Spatial-Spectral unification. This specific ordering enables effective feature-level fusion of multi-temporal images while maintaining the spatial fidelity required for dense prediction tasks.

---

> ### Author Response · Authors · 2025-11-30
> **Response to Wnht (2/5)**
>
> > **[W3 & Q3 & Q4]**: Despite the generalist formulation and architectural adaptability through hypernetworks to create adapted linear layers ... What are the key differences between the proposed approach and a foundation model ... Since the proposed method depends on input and output dimensions
>
> **[A3]** There appears to be a misunderstanding regarding our dimensional unification mechanism. We clarify that STSUN **can** accommodate different dimensions (e.g. shorter time series, varying spectral bands) **without** structural modification or retraining.
>
> The **Dimension Unification Module** does not use static linear layers. Instead, it employs a Hypernetwork that generates **adaptive mapping weights** conditioned on the input metadata. Taking the input spectral dimension unification shown on the left side of Figure 6(a) as an example:
>
> - **Mechanism:** Consider mapping an input image of shape (H_1, W_1, C_1) to a unified feature (H_1 × W_1, C). The input metadata is the spectral wavelength vector of size (C_1).
>
> - **Dynamic Generation:** Even if C_1 changes (e.g., from 3 bands to 13 bands), the metadata vector changes accordingly. The fixed Hypernetwork takes this variable metadata and generates adaptive weights W with shape (C_1, C) and bias B with shape (C).
>
> - **Application:** These generated W and B effectively form a linear layer that map the input image to unified feature. Therefore, the "linear layer" adapts dynamically to any input dimension defined by the metadata, allowing immediate generalization across datasets without manual adjustment.
>
>
> > **[W4]**: It is unclear why foundation models, essential methods in remote sensing nowadays ...
>
> **[A4]** We thank the reviewer for the suggestion. The core contribution of this work is the **Spatial-Temporal-Spectral Unified Strategy**. Our experiments focus on demonstrating that a model based on this strategy can unify the dimensions of input/output data, thereby achieving unification across data, tasks, and categories.
>
> We did not initially include foundation models because our core focus differs from that of FMs, and comparing fully supervised models trained on limited data) with FMs (pre-trained on massive datasets) is inherently unfair. We address your specific concerns below:
>
> **1. Response to Point (a): Difference in Research Goals** Remote sensing FMs primarily aim to learn robust representations for general downstream tasks, but they typically do not address the unification of input/output dimensions.
>
> - Most FMs are rigid regarding input modalities. For example, RingMo is designed for RGB images, S2MAE for 13-band multispectral images.
>
> - In contrast, **STSUN** is designed to accept inputs with **arbitrary** spatial sizes, temporal lengths, and spectral bands within a single model.
>
> - Therefore, our work addresses a different research direction (dimensional and architectural unification) compared to FMs (representation learning).
>
>
> **2. Response to Points (b) & (c): Fairness of Comparison** A fair comparison depends not only on parameter size or output heads but critically on **training paradigms and data scale**. FMs leverage massive-scale pre-training followed by fine-tuning, whereas STSUN is trained from scratch using only the downstream task data. Comparing a supervised model directly against pre-trained models is unfair and methodologically unreasonable.
>
> **3. Empirical Comparison with Foundation Models** Despite the aforementioned unfairness, we conducted additional experiments to demonstrate that STSUN can match or surpass FMs. We compared STSUN against several well-known FMs on the **LEVIR-CD** (building change detection) and **DynamicEarthNet** (LULC) datasets.
>
> - **Settings:** Since the compared FMs only support mono-temporal inputs, STSUN performs semantic segmentation using single images on the long-term DynamicEarthNet dataset for a direct comparison.
>
> - **Results:** As shown in the table below, on **LEVIR-CD**, STSUN surpasses many FMs and achieves performance comparable to RingMo and CMID. On **DynamicEarthNet**, STSUN outperforms all compared FMs, achieving SOTA results.
>
>
> | **Model** | **LEVIR-CD (F1-score %)** | **DynamicEarthNet (mIoU %)** |
> | --- | --- | --- |
> | Seco [4] | 90.14 | —   |
> | SatMAE [5] | 87.65 | 39.9 |
> | RingMo [6] | 91.86 | —   |
> | SSL4EO [7] | 89.05 | 42.1 |
> | CMID [8] | 91.72 | 43.5 |
> | SatLas [9] | 90.62 | 40.7 |
> | Scale-MAE [10] | —   | 41.7 |
> | SkySense [11] | —   | 46.5 |
> | **STSUN (Ours)** | **91.59** | **54.9** |

---

> ### Author Response · Authors · 2025-11-30
> **Response to Wnht (3/5)**
>
> > **[W5 & Q5]**: There is insufficient comparison with Perceiver IO ... What are the differences between the proposed method and Perceiver IO
>
> **[A5]** Thank you for this suggestion. We would like to clarify the fundamental differences between our approach and Perceiver IO, which make a direct comparison unsuitable for the specific challenges of remote sensing (RS) unification.
>
> 1. **Fundamental Mechanism Difference:** While both methods aim for unified representations, the mechanisms differ significantly. Perceiver IO utilizes cross-attention to map input data to a fixed latent array and subsequently maps this latent array to outputs. In contrast, our STSUN utilizes **hypernetworks** conditioned on dimensional metadata. This generates adaptive mapping networks dynamically, allowing for a flexible transformation from diverse inputs to a unified representation, and from that unified representation to diverse outputs. This realizes true Data-Task-Category unification.
>
> 2. **Incompatibility with RS Spectral Diversity (Input):** Perceiver IO handles different input modalities by concatenating a learned position/modality embedding to the input. In RS, spectral bands have specific physical meanings and vary significantly (e.g., RGB vs. Near-Infrared/Shortwave-Infrared). Each band combination effectively acts as a distinct "modality." Applying Perceiver IO would require training specific embeddings and linear layers for every possible band combination, which is computationally expensive and rigid.
>
> 3. **Incompatibility with Dynamic Class Sets (Output):** Similarly, for the output channel dimension (semantic categories), Perceiver IO would require specific trainable embeddings and projection layers for every distinct set of target classes.
>
> 4. **Unification Mechanism in STSUN:** In contrast, STSUN employs a single hypernetwork that takes spectral or category metadata as input to generate the weights for the mapping layers. This allows our fixed hypernetwork to adapt to *any* arbitrary combination of input spectral bands or output semantic categories without retraining or adding parameters.
>
>
> > **[W6]**: The proposed Local-Global Window mechanism is not novel ...
>
> **[A6]** We respectfully clarify that our Local-Global Window Attention (LGWA) is distinct from the mechanism in Swin Transformers.
>
> Swin Transformer replaces standard global attention with **shifted window attention** to reduce computation; it is inherently a local attention mechanism that approximates global context over multiple layers. In contrast, our LGWA explicitly integrates **Global Attention** with **Multi-shape Local Attention**. We design multiple local window shapes to capture features in various directions while simultaneously maintaining a global branch. This allows the model to adapt effectively to RS data with varying spatial resolutions and coverage. To validate this, we included Swin's window attention in our ablation study. The results demonstrate that LGWA outperforms standard window attention, proving the effectiveness of our hybrid design.
>
> > **[W7]**: There is no LULC scenario with time series ...
>
> **[A7]** We apologize if this was not sufficiently highlighted. Our experiments on LULC do include a time-series scenario. Specifically, we utilized the DynamicEarthNet dataset, which consists of time series with a length of 24 steps spanning a 2-year period. This is stated in Section 4 (dataset introduction) and Table 1. The results on DynamicEarthNet demonstrate STSUN's effectiveness in long-term temporal dense prediction tasks.
>
> > **[W8]**: STSUN appears to be the only model trained on six datasets ...
>
> **[A8]** We appreciate the opportunity to clarify our experimental setting.
>
> **1. On Fairness of Comparison** To ensure fairness, the experimental resutls included the **STSUN_single** variant (trained on a single dataset like competing methods) alongside the multi-dataset STSUN.
>
> - **Validity:** Results show that `STSUN_single` achieves performance comparable to SOTA methods trained on identical data, verifying the effectiveness of our proposed architecture.
>
> - **Unification Benefit:** The standard `STSUN` (trained on multiple datasets) outperforms `STSUN_single` and competing methods. This validates our core hypothesis: the **Data-Task-Category unification strategy** effectively leverages diverse data to enhance model performance.
>
>
> **2. On Additional Comparisons**
>
> - **TSViT:** We have already compared STSUN with TSViT on the **DynamicEarthNet** dataset (which focuses on long-term dense prediction), where STSUN demonstrated superior performance.
>
> - **Foundation Models:** As detailed in **[A4]**, comparing our supervised method against FMs fine-tuned on each dataset is inherently unfair due to the vast difference in pre-training data volume. However, even under this disadvantageous setting, STSUN demonstrates competitive performance , as shown in **[A4]**.

---

> ### Author Response · Authors · 2025-11-30
> **Response to Wnht (4/5)**
>
> > **[W9]**: The experiments lack standard error estimates evaluating the variability of the proposed method ...
>
> **[A9]** Thank you for your advice. We have conducted repeated experiments using 10 random initialization seeds for the category unification ablation study (Table 10). The mean and standard error results are presented below:
>
> | **Model** | **LoveDA Dataset (mIoU)** | **Dynamic Dataset (mIoU)** |
> | --- | --- | --- |
> | **STSUN_single** | 65.61 ± 0.07 | 74.68 ± 0.04 |
> | **STSUN_fixed** | 64.64 ± 0.11 | 73.98 ± 0.08 |
> | **STSUN_flexible** | **65.76 ± 0.08** | **74.84 ± 0.05** |
>
> The results confirm that `STSUN_flexible` (our proposed strategy) performs significantly better than the `STSUN_fixed` baseline and achieves parity with (or slight improvements over) the single-task `STSUN_single` model, proving the effectiveness and stability of our category unification strategy.
>
> > **[W10]**: There is a lack of ablation study to justify the selection of all proposed modules.
>
> **[A10]** We have provided comprehensive ablation studies for our core contributions: the Spatial-Temporal-Spectral Unified Strategy and the LGWA mechanism.
>
> - **Dimensional Unification Order:** Ablated in **Section 4.5**.
>
> - **Data Unification:** Ablated in **Section A.7**.
>
> - **Task Unification:** Ablated in **Section 4.3**.
>
> - **Category Unification:** Ablated in **Section 4.4**.
>
> - **LGWA:** Ablated in **Section 4.5**, demonstrating its superiority over global-only and window-only attention mechanisms.
>
>
> > **[W11 & C1 & C2]**: The overall paper is difficult to read due to the introduction of numerous proposed concepts ... Section 3.2 lacks references to the architecture details of each module ... Figure 6: this crucial figure is confusing
>
> **[A11]** Thank you for your valuable feedback on the presentation. We have revised the manuscript to improve readability significantly:
>
> 1. **Simplified Notation:** We have optimized the naming conventions for modules and layers to reduce redundancy and improve flow.
>
> 2. **Clarified References:** We verified and corrected the cross-references in **Section 3.2** regarding **Section A.1** and **Figure 6** to ensure details are easy to locate.
>
> 3. **Improved Figure 6:** We have redesigned Figure 6 to clearly illustrate the interaction between modules and the specific dimension matching at each stage.
>
> 4. **Math & Symbols:** We have thoroughly proofread all formulas and symbols to ensure they are both mathematically rigorous and easy to follow.
>
>
> > **[Q1]**: Considering the details provided in W.1, to what extent does unifying dense tasks provide benefits ...
>
> **[A12]** In Section 4.3, we conducted a comprehensive ablation study specifically targeting task unification. The experimental results demonstrate that the task-unified model (STSUN), trained across multiple datasets, significantly outperforms the task-specific model trained on a single dataset (STSUN_single). Not only can STSUN be directly applied to multiple tasks and datasets, but it also achieves superior performance compared to the single-task baseline.
>
> This finding aligns with existing literature, which suggests that learning correlations across multiple tasks can enhance the performance of foundation models on downstream tasks [1] or improve the robustness of the model's representation [2].
>
> > **[Q2]**: Considering the details provided in W.1 and W.4, to what extent does a fixed set of classes introduce ...
>
> **[A13]** In Section 4.4, we investigated category unification through ablation studies. The results indicate that the flexible prediction model using category embeddings (STSUN_flexible) outperforms the model using a fixed set of classes (STSUN_fixed). Crucially, STSUN_flexible can perform predictions for multiple category sets simultaneously. It maintains high prediction performance while significantly boosting the efficiency of downstream category prediction.
>
> A similar approach is observed in recent works like CLIP [3], which encodes category text into embeddings to guide the model toward flexible category prediction. These experimental evidence confirms that utilizing category embedding sets allows the model to predict across diverse category sets without compromising the backbone's performance or generalization capabilities, thereby enhancing downstream efficiency.

---

> ### Author Response · Authors · 2025-11-30
> **Response to Wnht (5/5)**
>
> > **[Q6]**: How are the category embedding set and selected category subsets defined ...
>
> **[A14]** We address these points individually below:
>
> 1. **Category Embedding Set:** This is defined as a comprehensive collection of embeddings, where a predefined, trainable embedding is assigned to every category included across multiple remote sensing scenes.
>
> 2. **Selected Category Subset:** This is defined as the specific subset of embeddings extracted from the global set. The subset corresponds to the categories present in the target remote sensing scene.
>
> 3. **Limitation on Categories:** No, the method is **not** limited to a fixed number of categories. For a category set encompassing multiple remote sensing scenes, our method allows for the selection of an arbitrary number of categories for prediction, independent of the total size of the category set. This flexibility is achieved because the category metadata (i.e., the category embeddings) changes adaptively based on the target categories during inference. This mechanism effectively guides the model to perform adaptive category prediction for any given number of classes.
>
>
> > **[C3]**: Figure 6: to what corresponds to the list of numbers related to the spectral wavelength input in (a)?
>
> **[A15]** The list of numbers in Figure 6(a) corresponds to the spectral wavelengths of the input remote sensing image. Specifically, the example illustrates a list of 13 wavelengths corresponding to Sentinel-2 imagery, with the unit measured in micrometers (μm).
>
> > **[C4]**: Table 1: please explain all acronyms in the caption.
>
> **[A16]** Thank you for the suggestion. We have revised the caption of Table 1 to include full explanations for all acronyms used.
>
> [1] Zheng, Z., Lv, L., He, J., & Zhang, L. (2025). UniRS: Towards Unified Multi-task Fine-Tuning for Remote Sensing Foundation Model. IEEE Transactions on Geoscience and Remote Sensing.
>
> [2] Wang, D., Zhang, J., Xu, M., Liu, L., Wang, D., Gao, E., ... & Zhang, L. (2024). MTP: Advancing remote sensing foundation model via multitask pretraining. IEEE Journal of Selected Topics in Applied Earth Observations and Remote Sensing, 17, 11632-11654.
>
> [3] An, K., Wang, Y., & Chen, L. (2025). Soft-Guided Open-Vocabulary Semantic Segmentation of Remote Sensing Images. IEEE Transactions on Geoscience and Remote Sensing.
>
> [4] Manas, O., Lacoste, A., Giró-i-Nieto, X., Vazquez, D., & Rodriguez, P. (2021). Seasonal contrast: Unsupervised pre-training from uncurated remote sensing data. In *Proceedings of the IEEE/CVF International Conference on Computer Vision* (pp. 9414-9423).
>
> [5] Cong, Y., Khanna, S., Meng, C., Liu, P., Rozi, E., He, Y., ... & Ermon, S. (2022). Satmae: Pre-training transformers for temporal and multi-spectral satellite imagery. *Advances in Neural Information Processing Systems*, *35*, 197-211.
>
> [6] Sun, X., Wang, P., Lu, W., Zhu, Z., Lu, X., He, Q., ... & Fu, K. (2022). RingMo: A remote sensing foundation model with masked image modeling. *IEEE Transactions on Geoscience and Remote Sensing*, *61*, 1-22.
>
> [7] Wang, Y., Braham, N. A. A., Xiong, Z., Liu, C., Albrecht, C. M., & Zhu, X. X. (2023). SSL4EO-S12: A large-scale multimodal, multitemporal dataset for self-supervised learning in Earth observation [Software and Data Sets]. *IEEE Geoscience and Remote Sensing Magazine*, *11*(3), 98-106.
>
> [8] Muhtar, D., Zhang, X., Xiao, P., Li, Z., & Gu, F. (2023). Cmid: A unified self-supervised learning framework for remote sensing image understanding. *IEEE Transactions on Geoscience and Remote Sensing*, *61*, 1-17.
>
> [9] Bastani, F., Wolters, P., Gupta, R., Ferdinando, J., & Kembhavi, A. (2023). Satlaspretrain: A large-scale dataset for remote sensing image understanding. In *Proceedings of the IEEE/CVF International Conference on Computer Vision* (pp. 16772-16782).
>
> [10] Reed, C. J., Gupta, R., Li, S., Brockman, S., Funk, C., Clipp, B., ... & Darrell, T. (2023). Scale-mae: A scale-aware masked autoencoder for multiscale geospatial representation learning. In *Proceedings of the IEEE/CVF International Conference on Computer Vision* (pp. 4088-4099).
>
> [11] Guo, X., Lao, J., Dang, B., Zhang, Y., Yu, L., Ru, L., ... & Li, Y. (2024). Skysense: A multi-modal remote sensing foundation model towards universal interpretation for earth observation imagery. In *Proceedings of the IEEE/CVF Conference on Computer Vision and Pattern Recognition* (pp. 27672-27683).

---

### Note · Authors · 2026-01-27

I have read and agree with the venue's withdrawal policy on behalf of myself and my co-authors.

---

### Meta-Review · Area_Chair_nF7U · 2026-01-05

**Summary:**

This paper proposes a framework for reote sensing dense prediction and seeks to address heterogeneous factors/conditions between different images. The core idea involves a spatial-spectral-spectral module that claims to exploit spectral continuity and spatial promiximity, followed by a shared-weight encoder that processes each temporal image independently. Next, the temporal unified module provides a bridge between the encoder and decoder by "unifying" the temporal dimension. A shared decoder refines the fused representation, which is then followed by the OSSUM that undertakes the final mapping. Overall, the reviewers found that the problem is interesting, the strategy is reasonable and the experimental results are convincing. However, several concerns were raised, including limited novelty, positioning this in the context of foundation-model work, ablation studies etc. Out of these, comments pertaining to novelty are specifically relevant and important in the context of the bar required for an ICLR paper. This paper is an interesting engineering approach to solve a problem commonly encountered in remote sensing (and that a plethora of papers are addressing). The approach proposed here is reasonable, but from the perspective of new algorithmic/theoretical insights, the contribution is limited. When reviewing the reviewer feedback/response and the manuscript, I also noticed concerns in the overall formulation/assumption - for instance, the authors in a few places mention spectral continuity. However, spectral continuity is only typical in hyperspectral imagery - infact, in multispectral imagery, spectral continuity is not guaranteed at all - bands can and will often have gaps, their FWHM bandwidths may be different etc.

**Reviewer Concerns:**

The authors have done a good job in addressing several of the comments and feedback from the reviewers. Concerns pertaining to the novelty however stand. Concerns pertaining to positioning this relative to foundation models also stand - there are a lot of emerging works that can use foundation models such as Prithvi and deploy these with minimal adaptation to hyperspectral, multispectral and multi-temporal imagery.

**Reviewer Scores:**

Based on the concerns, my conjecture is that the scores would stand.

---

### Decision · Program_Chairs · 2026-01-26

Reject